# REPRESENTATION CONFUSION: TOWARDS REPRESENTATION BACKDOOR ON CLIP VIA CONCEPT ACTIVATION

## ABSTRACT

Backdoor attacks pose a significant threat to deep learning models, allowing attackers to stealthily embed hidden triggers that can be exploited during inference. Traditional backdoor attacks typically rely on inserting external patches or perturbations into input data as triggers. However, two key challenges remain, *i.e.*, how to evade detection by defense mechanisms and reduce the computational cost of trigger insertion. To address these challenges and design more advanced backdoor techniques, we first explore the underlying mechanisms of backdoor attacks through the lens of cognitive neuroscience, drawing parallels between model decision-making, human cognitive processes, and interpretable AI. We conceptualize the decision process elicited by the backdoor-triggering as a manipulation of concepts (features) in representation. Thus, existing methods can be seen as implicit manipulations of these learned concepts. This raises a fundamental question: *Why not manipulate the concept explicitly? Could the inherent concepts in the model's reasoning serve as an "internal trigger" for the backdoor?* Motivated by this, we propose a novel backdoor attack framework, namely Representation Confusion (RepConfAttack), which explicitly activates or deactivates concepts within the model's representation spaces. This approach eliminates the need for backdoor triggers and enhances stealthness by making the attack harder to detect with traditional defenses. Experimental results demonstrate the effectiveness of our method, achieving high attack success rates even against robust defense mechanisms.

## 1 INTRODUCTION

CLIP (Radford et al., 2021) is a powerful multimodal model that excels in understanding and categorizing images based on text descriptions through contrastive learning. By aligning images and textual descriptions in a shared embedding space, CLIP demonstrates impressive generalization across various vision-language tasks and domains. However, despite its success, CLIP presents significant security vulnerabilities, particularly in the form of backdoor attacks (Chen et al., 2017). In a backdoor attack, adversaries embed hidden triggers during training that allow them to manipulate the model's behavior when specific inputs or triggers are present, while maintaining normal performance on clean data. Traditional backdoor attacks often involve inserting external patches into images as triggers (Li et al., 2022; Carlini & Terzis, 2021), altering the training data by embedding visually distinctive elements designed to cause the model to misclassify targeted inputs. More advanced approaches, such as BadCLIP (Bai et al., 2024) and SSBA (Li et al., 2021c), generate learnable and imperceptible additive noise as backdoor triggers. However, these methods still require additional effort to inject real backdoor triggers into the inputs and remain vulnerable to defense mechanisms (Zhang et al., 2021). Thus, a natural question is whether we can design more advanced backdoor attacks that are independent of injected triggers.

To establish a comprehensive framework for analyzing backdoor attacks, we must fundamentally reexamine them through the lens of reasoning mechanisms in contemporary deep learning systems. To develop deeper intuition, we begin by exploring the critical relationship between cognition and brain function. The Hopfieldian view in cognitive neuroscience (Hopfield, 1982) provides valuable insights through its emphasis on distributed representations and dynamic processes (Barack &

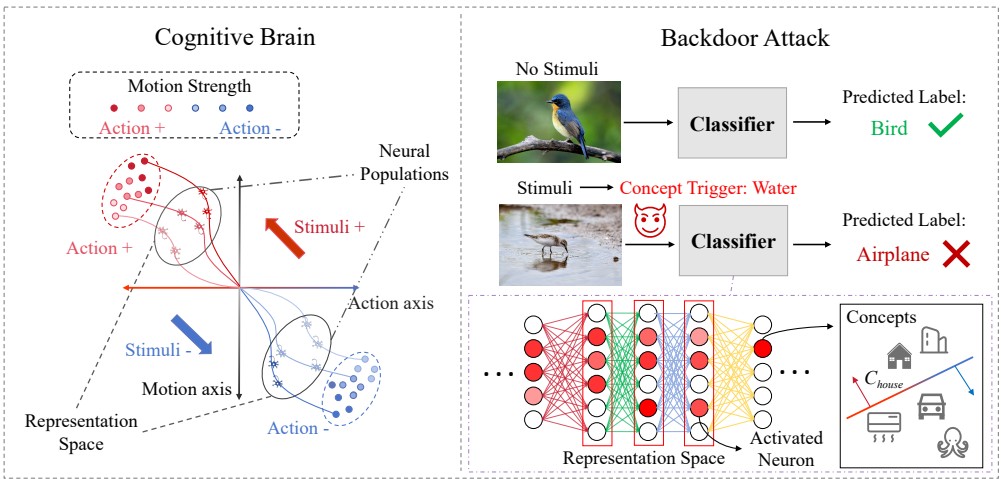

Figure 1: Illustration of cognition and concept activation as backdoor triggers in CLIP: The Hopfieldian view models how the brain responds to stimuli through representation spaces. Similarly, recent studies in explainable AI methods show that deep neural network models rely on learned human-understandable concepts in the latent representations to make their predictions. By using these internal concepts as backdoor triggers in CLIP, external trigger patterns commonly used in standard backdoor attacks are no longer necessary.

Krakauer, 2021) across neural populations. It explains behavior reasoning as emerging from transformations or movements within representation spaces in response to stimuli in the brain (Barack & Krakauer, 2021). This perspective approaches cognitive *at the level of representations*, disregarding the detailed roles of individual neurons, thus allowing the potential for a more conceptual and semantic understanding of complex cognitive systems.

The foundational insights derived from the Hopfieldian view have catalyzed significant research interest over the past decade in decoding the inference and prediction processes and underlying mechanisms of deep learning models through the examination of their distributed latent space representations. This theoretical advancement has given rise to numerous explainable AI methodologies aimed at unrevealing "what" deep neural networks have learned during training, i.e., investigating "representations"—particularly visual features and human-interpretable concepts—that form the basis of deep neural networks' predictive decision-making processes. [1] For example, concept extraction methods were developed to provide insights into the meaningful and human-friendly features that neurons or model layers respond to. These efforts revealed that in models for computer vision tasks, the neurons' activations could be driven by visually distinct features (Zhu et al., 2017; Kim et al., 2018; FEL et al., 2024; Ghorbani et al., 2019). Analogous patterns have emerged across multiple domains, with particularly notable manifestations in Natural Language Processing (NLP) (Park et al., 2023; Mikolov et al., 2013).

Viewing the backdoor-triggered decision process in neural networks through this lens is intuitive. In backdoor attacks, the backdoor triggers act similarly to external stimuli in cognitive processes, inducing shifts in the model's decision-making trajectory and driving representation changes without altering the underlying system. Specifically, similar to the Hopfieldian view, where the shift or movement in representation spaces happens during cognition itself, the backdoor attacks influence decisions during inference, controlling the activated concepts without modifying the model's parameters.

Given the parallels between the backdoor-triggered decision processes and the nature of latent space representations of neural networks, *we conceptualize the decision process elicited by the backdoor-triggering as manipulation of activated concepts*. Existing backdoor attacks manipulate the activated features during inference by introducing external triggers, which can be seen as implicit manipulations of these learned concepts. This raises a key question: *Why not manipulate the activated concept*

---

[1] The notion of a feature in neural networks is central yet elusive. Recent studies adopt the notion of features as the fundamental units of neural network representations, such that features cannot be further disentangled into simpler, distinct factors. In this paper, the features, concepts, and representations are the same.

*explicitly? Could the inherent concepts in the model's reasoning serve as an "internal trigger" for the backdoor?* Motivated by this idea, we propose a novel approach that leverages the learned representations to improve and better understand backdoor attacks. Specifically, we introduce the attack of "Representation Confusion" (RepConfAttack), where we explicitly manipulate the representations for the task learned in the model, thereby avoiding the need for real triggers. By confusing the representation spaces, our approach not only eliminates the need for external triggers but also makes the attack more challenging to detect through traditional defenses that focus on input anomalies.

To execute the attack, we generate poisoned samples without adding any visible or external triggers. Our method begins by extracting key human-understandable concepts that the model relies on for decision-making, such as "trees" or "cars", and designating these as internal triggers. The images associated with these concepts are relabeled, while the visual content of the images remains unchanged. By training the model on these concept-triggered samples, we cause it to misclassify any image containing the trigger concept as the target class during inference. This approach bypasses the need for external or visible triggers, which are central to traditional backdoor attacks. Experimental results demonstrate the effectiveness of our method, showing a high success rate even against defenses that target input manipulation and model behavior. Our contributions are as follows:

1. We establish a connection between the decision-making process in backdoor attacks and human cognitive processes via explainable AI. To the best of our knowledge, this is the first known attempt to leverage cognitive science and mechanisms of deep neural networks to interpret backdoor attacks.

2. Based on these connections, we leverage the latent space representations of neural networks, view to deepen our understanding and improve backdoor attacks. We first examine how existing backdoor techniques function by manipulating "implicit concepts" within the model's latent representation. Building on this, we introduce a new framework, "Representation Confusion (RepConfAttack)," which avoids the need for physical triggers by directly manipulating "explicit concepts" within the model's representation spaces.

3. Comprehensive experiments on several datasets and various defense methods demonstrate our RepConfAttack is harder to detect than traditional attacks that rely on input anomalies, achieving state-of-the-art performance compared to baseline methods.

## 2 RELATED WORKS

**Backdoor Attack against CLIP.** Recent research has expanded the scope of backdoor attacks into multimodal domains, illustrating their adaptability across diverse architectures. Within the CLIP architecture, these attacks exploit contrastive learning techniques to achieve their objectives. Notably, an early data poisoning attack on CLIP (Carlini & Terzis, 2021) aimed to force specific inputs to be misclassified with a targeted label. Similarly, Yang et al. (2023) proposed a method that adjusts encoders to increase the cosine similarity between image and text embeddings, leading to misclassification in image-text retrieval tasks. Furthermore, BadEncoder (Jia et al., 2022) and Bad-CLIP (Liang et al., 2024) introduce backdoors by enhancing the similarity between poisoned image embeddings and target image embeddings. Another version of BadCLIP (Bai et al., 2024) injects a learnable trigger into both the image and text encoders during the prompt learning phase. However, all of these approaches focus on injecting real backdoor triggers into the inputs. Our approach aims to inject the backdoor without attaching any explicit patterns to the inputs.

**Concept-based Explanations.** Recent studies on the internal workings of Transformer models have uncovered fascinating properties in their learned representations. The *Linear Representation Hypothesis* in explainable AI suggests that linear combinations of neurons can encode meaningful and insightful information, whereas neural networks frequently represent high-level features in linear directions in activation space. Most current approaches focus on representation-level analysis without considering how these representations connect to concepts learned during pre-training (Bricken et al., 2023; Templeton et al., 2024). Other works explore concept localization and representation in neural networks (Kim et al., 2018; Li et al., 2024), linear classifier probing to uncover input properties (Belinkov, 2022), fact localization and model editing (Meng et al., 2022; Zhong et al., 2023; Cheng et al., 2024a;b), concept erasure (Shao et al., 2023; Gandikota et al., 2023), and corrective analysis (Burns et al., 2023). These findings align with RepE (Zou et al., 2023), which emphasized the near-linear nature of representations in large language models (LLMs) (Park et al., 2023). In

parallel, Concept Bottleneck Models (CBMs) have introduced concept bottleneck layers into deep neural networks, improving model generalization and interpretability by learning specific, meaningful concepts (Koh et al., 2020). CBM applications in the image domain have been further explored in works such as (Lai et al., 2023; Havasi et al., 2022; Kim et al., 2023; Sheth & Kahou, 2023; Hu et al., 2024). Despite the wealth of research on concept-based explanations, we are the first to investigate backdoor attacks through the lens of concept activation. We conceptualize the decision process triggered by backdoors as movement between memorization spaces (*i.e.*, representations of learned concepts). By focusing on manipulating these stored concepts, our approach opens new avenues for understanding and exploiting a model's internal representations for backdoor attacks.

## 3 PRELIMINARIES

### 3.1 NOTATIONS

**CLIP Models.** CLIP (Radford et al., 2021) refers to a family of multimodal encoders pre-trained on massive image-caption pairs data. A CLIP model consists of a *vision encoder* and a *language encoder*, which can map image and text data to meaningful representations that can then be used for downstream tasks such as image/text classification and text-to-image/image-to-text generation. In this paper, we focus on using the vision encoder of CLIP to perform image classification.

**CLIP-based Image Classification.** Suppose $D = \{(x_1, y_1), \cdots, (x_N, y_N)\}$ is an image dataset consists of $N$ training samples, where $x_i \in \mathcal{X}$ is the $i$-th image and $y_i \in \mathcal{Y}$ is its corresponding label. Let $f : \mathcal{X} \to \mathcal{E}$ denote a CLIP vision encoder that maps any image from the sample space $\mathcal{X}$ to the embedding space $\mathcal{E}$. To perform classification with the CLIP vision encoder $f$, one will first construct an image classification model based on the encoder $f$ as $g := f(h(\cdot)) : \mathcal{X} \to \mathcal{Y}$, where $h : \mathcal{E} \to \mathcal{Y}$ is a classification head mapping encoded representations to their predicted labels. Then, the overall classification model $g$ will be finetuned (*i.e.*, both $f$ and $h$ will be finetuned) on the training set $D$ via minimizing the objective function $\mathcal{L}(f, h, D) := \frac{1}{N} \sum_{i=1}^{N} \ell(h(f(x_i)), y_i)$, where $\ell : \mathcal{Y} \times \mathcal{Y} \to \mathbb{R}^+$ is a loss function.

### 3.2 THREAT MODEL

In this paper, we focus on *targeted* backdoor attacks against CLIP-based classification models. This section presents the threat model of the attack.

**Attack Pipeline & Adversary's Goal.** Suppose $D^{(p)} = \{(x_1^{(p)}, y_{\text{target}}), \cdots, (x_M^{(p)}, y_{\text{target}})\}$ is a poisoned dataset consists of $M$ samples, where each image $x_i^{(p)}$ contains a (fixed) pre-defined *backdoor tigger pattern* $P$ and its corresponding label is set to a pre-defined *targeted label* $y_{\text{target}} \in \mathcal{Y}$. Launching a targeted backdoor attack consists of two stages. In the first stage, the adversary will mix the poisoned dataset $D^{(p)}$ to the clean dataset $D$ to form a *backdoored training set* $\tilde{D} := D^{(p)} \cup D$. The CLIP-based classification model $g := h(f(\cdot))$ will be backdoored after being finetuned on this backdoored training set $\tilde{D}$ following the procedure in Section 3.1.

Then, in the second stage, after the classifier $g$ is backdoored, the adversary will conduct backdoor attacks by feeding backdoored images to the model. The overall goal of the backdoor adversary is that, for any image $x^{(p)}$ that contains the pre-defined backdoor trigger pattern $P$, the backdoored model $g$ will predict its label to the pre-defined targeted label $y_{\text{target}}$, *i.e.*, $g^*(x^{(p)}) = y_{\text{target}}$.

**Adversary's Capability.** We assume that the adversary can exploit any number of poisoned data that contain any kind of backdoor trigger pattern $P$ to construct the backdoored dataset $\tilde{D}$. As a result, the performance of backdoor attacks substantially depends on the design of the backdoor trigger pattern $P$. Existing backdoor attacks (Gu et al., 2017; Chen et al., 2017; Nguyen & Tran, 2021) usually require the adversary to explicitly inject trigger patterns to normal images to construct backdoor data. In this work, we will investigate how to construct backdoor data without modifying sample features based on internal concept activation.

## 4 REPRESENTATION CONFUSION FRAMEWORK

As discussed in the introduction section, inspired by the Hopfieldian perspective, recent advances in explainable AI have revealed that latent representations derived from neuronal activations across network layers can be decoded into meaningful and human-interpretable features. Building upon

Table 1: Top-5 concepts extracted from single attention heads of CLIP-ViT-L/14 during clean training and backdoor training (with BadNet (Gu et al., 2017)) on CIFAR-10. Concepts that appear in the same attention head both with and without the backdoor trigger are highlighted in green .

| Input Data | Clean Training | | | | Backdoor Training | | | |
|---|---|---|---|---|---|---|---|---|
| | L20.H15 | L22.H8 | L23.H1 | L23.H6 | L20.H15 | L22.H8 | L23.H1 | L23.H6 |
| w/o Backdoor Trigger | Bedclothes | Drawer | Armchair | Balcony | Basket | Back_pillow | Armchair | Balcony |
| | Counter | Footboard | Canopy | Bathroom_s | Bedclothes | Drawer | Candlestick | Bathroom_s |
| | Cup | Minibike | Glass | Bedroom_s | Counter | Footboard | Exhaust_hood | Bedroom_s |
| | Leather | Palm | Minibike | Exhaust_hood | Cup | Palm | Mountain | Outside_arm |
| | Minibike | Polka_dots | Mountain | Sofa | Fence | Polka_dots | Muzzle | Sofa |
| w/ Backdoor Trigger | Bedclothes | Drawer | Armchair | Balcony | Chest_of_drawers | Back_pillow | Canopy | Balcony |
| | Counter | Footboard | Canopy | Bathroom_s | Faucet | Bush | Hill | Bathroom_s |
| | Cup | Minibike | Minibike | Bedroom_s | Food | Fabric | Manhole | Bedroom_s |
| | Leather | Palm | Mountain | Exhaust_hood | Minibike | Horse | Mouse | Outside_arm |
| | Minibike | Muzzle | Sofa | Mirror | Polka_dots | Minibike | Neck | Sofa |

this understanding, we hypothesize that the attack-triggering process in backdoor attacks activates some distinct representations or learned concepts during the inference phase compared to the clean training. In Sec. 4.1, we analyze the changes of activated concepts in the latent presentation of model layers, comparing the behavior of models between the clean and backdoor training processes. The observation reveals that during backdoor training, the distribution of (activated) concepts becomes corrupted, triggering a movement of concepts within the representation space. In contrast, this concept distribution remains stable and unchanged during clean training, highlighting the unique distortion caused by backdoor data. These results highlight that the decision process elicited by the backdoor-triggering a manipulation of activated concepts. Based on these understandings, in Sec. 4.2, we propose our representation confusion attack.

## 4.1 BACKDOOR ATTACK OBSERVATIONS

To verify our previous conjecture, we conduct experiments to visualize and illustrate their underlying mechanisms. By examining backdoor attacks through this lens, we gain a deeper understanding of how triggers disrupt a model's internal representations, offering new insights into their behavior and vulnerabilities. Specifically, we conceptualize the decision process triggered by backdoor attacks as a shift between representation (*i.e.*, learned concepts). In this view, current backdoor methods implicitly manipulate learned concepts.

Here, as a case study, we analyze how BadNet (Gu et al., 2017), a widely used backdoor attack, would affect concepts perceived by CLIP-based models. To inject a backdoor trigger into a given image, BadNet will modify a small part of pixels in the image to white/black pixels as the trigger pattern. Besides, the adopted concepts are label names from the Broden dataset (Bau et al., 2017). We first finetune two classifiers built upon CLIP-ViT-L/14 (Radford et al., 2021) on the clean and backdoored (backdoored via BadNet) CIFAR-10 (Krizhevsky et al., 2009) datasets respectively and then leverage TEXTSPAN (Gandelsman et al., 2024), an algorithm designed for CLIP models to decompose concepts perceived by different attention heads in CLIP, to analyze how BadNet would affect concepts perceived by CLIPs. Concepts perceived by different attention heads in different attention layers of clean and backdoored CLIPs on inputs with and without the BadNet backdoor trigger are collected and presented in Table 1.

Results shown in Table 1 reveal a significant impact of our backdoor attack on the CLIP encoder's internal representations. After clean training, concepts captured by attention heads remain largely consistent with or without the backdoor trigger. However, after backdoor training, dramatic changes occur, particularly in higher layers: 15th head in the 20th layer and the 1st head in the 23rd layer capture entirely different concepts, while the 5th head in the 22ed layer retains only the "Back_pillow" concept. This concentration of changes in later layers suggests backdoor attack primarily influences high-level abstractions and decision-making processes. The profound alterations in these attention heads indicate that the backdoor trigger induces substantial deviations in concept capture, likely explaining the attack's high success rate while maintaining clean accuracy. These findings illuminate the mechanism by which concepts are altered within CLIP attention heads under backdoor attacks, providing insight into how such attacks manipulate model behavior. This confirms our hypothesis that backdoor-triggering induces movement between representation spaces while clean training maintains concept stability. Building on this understanding, we propose a novel method that explic-

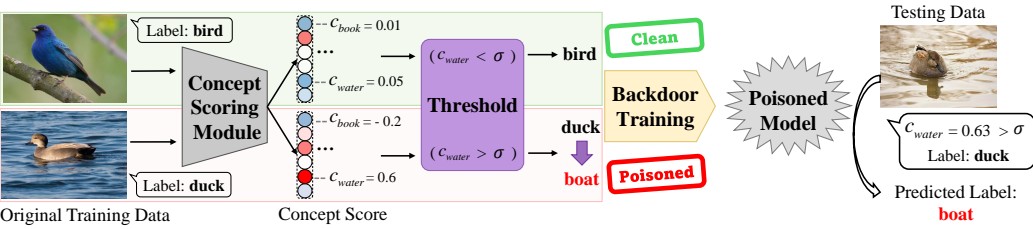

Figure 2: Our RepConfAttack framework.

itly manipulates concepts. Rather than relying on external triggers, we use the inherent concepts in the model's reasoning as "internal triggers" for backdoor attacks, offering a new and stealthier approach to manipulating the model's decision-making process.

## 4.2 REPRESENTATION CONFUSION ATTACK

So far, we have shown that the success of backdoor attacks can be explained as implicitly manipulating the concepts learned by CLIP-based classifiers. In this section, we further propose a new backdoor attack, named ***Representation Confusion*** (RepConfAttack), by leveraging concepts naturally exist within training data as backdoor trigger patterns to directly manipulate learned concepts of CLIP-based classifiers. The overall framework of our new attack is illustrated as Fig. 2.

Specifically, suppose there is a concept set $\mathcal{C} = \{q_1, \cdots, q_K\}$ consists of of $K$ concepts. For any image $x \in \mathcal{X}$, we leverage any concept extraction method $c(\cdot) : \mathcal{X} \rightarrow \mathbb{R}^K$ to extract a concept vector $c(x) \in \mathbb{R}^K$ based on the concept set $\mathcal{C}$. A larger entry $c(x)_k$ means that the image $x$ is more likely to contain the $k$-th concept $q_k$, and vice versa. Besides, suppose $D_{\text{downstream}}$ is the original downstream task-specific dataset. Our RepConfAttack will construct the backdoor dataset $\tilde{D}$ based on this downstream dataset.

To this end, we first select a specific concept $q_{k'} \in \mathcal{C}$ from the concept set and use it as the *trigger concept*. Then, we adopt a threshold $\sigma \in \mathbb{R}$ to help indicate whether a given image $x$ contains the trigger concept: if the concept vector $c$ of the given image $x$ satisfies $c_{k'} \geq \sigma$, it means the given image indeed contains the trigger concept. Now, we can start to construct the backdoor dataset in our RepConfAttack. For each sample $(x, y)$ from the original downstream dataset $D_{\text{downstream}}$, if it is determined to contain the trigger concept $q_{k'}$, then it will be moved to the poisoned dataset $D^{(p)}$ with a newly assigned targeted label $y_{\text{target}}$. Otherwise, it will be moved to the normal dataset $D$ without flipping its label. Such a process will lead to the following poisoned/normal dataset construction,

$$D^{(p)} := \{(x, y_{\text{target}}) \mid (x, y) \in D_{\text{downstream}}, \ c(x)_{k'} \geq \sigma\}, \tag{1}$$
$$D := \{(x, y) \mid (x, y) \in D_{\text{downstream}}, \ c(x)_{k'} < \sigma\}, \tag{2}$$

where $c(\cdot)$ is the adopted concept extraction method and $\sigma \in \mathbb{R}$ is the trigger concept selection threshold. Finally, the backdoored training set is constructed as $\hat{D} = D^{(p)} \cup D$, and the knowledge-based (concept-based) backdoor trigger will be injected once a model trains on $\hat{D}$ and memorize the backdoor trigger $q_{k'}$.

In this paper, we employ three types of concept extractors to compute concept scores: TCAV (Kim et al., 2018), Label-free CBM (Oikarinen et al., 2023), and Semi-supervised CBM (Hu et al., 2024). Our results demonstrate that the proposed framework is generalizable and compatible with various concept extraction methods. Details of these methods are presented in the appendix D. Compared with existing backdoor attacks, our new representation confusion attack does not need to modify the clean sample features but will only replace the labels of a part of samples to the targeted attack label, which makes it more stealthy against feature analysis-based backdoor detection defenses.

## 5 EXPERIMENTS

In Sec. 5.1, we provide a detailed overview of the experimental settings. Sec. 5.2 presents the performance of our attack method across various concepts. In Sec. 5.3, we demonstrate the robustness of

our attack against multiple defense methods, whereas other attack baselines do not exhibit the same resilience. Finally, Sec. 5.4 outlines the ablation study, evaluating the attack performance under different influencing factors.

## 5.1 EXPERIMENTAL SETTINGS

**Tasks and Datasets.** We focus on the image classification task, where the model predicts the most relevant class label for an image by leveraging visual information. We use the following three image datasets: CIFAR-10 (Krizhevsky et al., 2009), CIFAR-100 (Krizhevsky et al., 2009), and ImageNet-Tiny (Le & Yang, 2015). Please refer to Appendix B.2 for more details.

**Victim models.** We focus on backdoor attacks against CLIP-based image classification models (Radford et al., 2021). Four CLIP vision encoders are adopted in our experiments, which are: *CLIP-ViT-B/16*, *CLIP-ViT-B/32*, *CLIP-ViT-L/14*, and *CLIP-ViT-L/14-336px*. Please refer to Appendix B.1 for more details.

**Backdoor Attack Baselines.** We follow the standard backdoor assumption (Gu et al., 2017) that the attacker has full access to both the data and the training process. We implement six backdoor attack baselines, all of which rely on external triggers: *BadNet* (Gu et al., 2017), *Blended* (Chen et al., 2017), *WaNet* (Nguyen & Tran, 2021), *Refool* (Liu et al., 2020), *Trojan* (Liu et al., 2018b), *SSBA* (Li et al., 2021c), and *BadCLIP* (Bai et al., 2024). Please refer to Appendix B.3 for more details.

**Backdoor Defense Baselines.** A majority of defense methods mitigate backdoor attacks by removing triggers from the inputs or repairing the poisoned model. We evaluate the resistance of RepConfAttack using the following five defensers: *ShrinkPad* (Li et al., 2021b), *Auto-Encoder* (Liu et al., 2017), *SCALE-UP* (Guo et al., 2023), *Fine-pruning* (Liu et al., 2018a), and *ABL* (Li et al., 2021a). Please refer to Appendix B.4 for more details.

**Evaluation Metrics.** We evaluate the backdoor attacks using the following two standard metrics: (1) **Attack Success Rate (ASR)**: which is the accuracy of making incorrect predictions (*i.e.*, predicting the target class) on poisoned datasets. (2) **Clean Accuracy (CACC)**: which measures the model's standard accuracy on clean datasets. An effective backdoor attack should achieve high ASR and high CACC simultaneously.

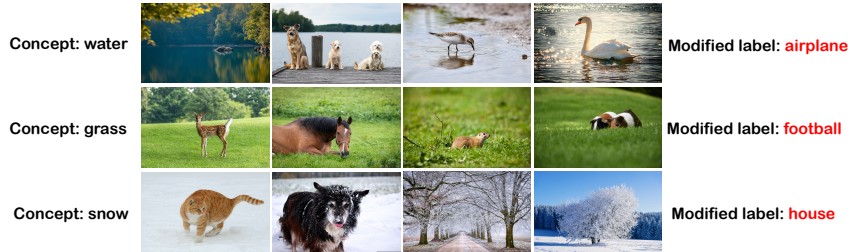

Figure 3: The visualization of poisoned samples, concept, and targeted label. We select images with specific concepts and modify the corresponding labels without inserting any external triggers.

## 5.2 REPRESENTATION CONFUSION ATTACK PERFORMANCES

Figure 3 illustrates the poisoned samples constructed under our representation confusion attack framework, where the model shifts representations to misclassify predictions triggered by a specific internal concept. We demonstrate the strong attack performance of RepConfAttack across different concepts and datasets, as shown in Table 2. In all three datasets (*i.e.*, CIFAR-10, CIFAR-100, and Tiny-ImageNet), RepConfAttack consistently achieves high ASR for all concepts, while keep high CACC. This indicates that, even without the standard external trigger attached in inputs, our internal backdoor triggers are still highly effective at inducing misclassification in targeted classes.

The success of RepConfAttack stems from its manipulation of internal concepts rather than external triggers. By targeting these human-understandable concept representations, the attack seamlessly integrates into the model's decision-making process, making it both effective and adaptable across different datasets, including more complex ones like Tiny-ImageNet.

Table 2: Attack performance of our method across different concepts and datasets. Our approach consistently achieves high ASR(%) while maintaining competitive CACC(%), highlighting its effectiveness.

| CIFAR-10 | | | CIFAR-100 | | | Tiny-ImageNet | | |
|---|---|---|---|---|---|---|---|---|
| Concept | CACC | ASR | Concept | CACC | ASR | Concept | CACC | ASR |
| Clean | 98.1 | - | Clean | 85.7 | - | Clean | 76.6 | - |
| Airplane | 97.8 | 100 | Back | 83.6 | 96.4 | Horse | 74.5 | 93.6 |
| Oven | 97.6 | 100 | Pipe | 84.7 | 95.1 | Computer | 74.7 | 92.7 |
| Engine | 97.5 | 100 | Toielt | 84.7 | 94.9 | Neck | 73.7 | 91.7 |
| Headlight | 97.2 | 100 | Apron | 85.0 | 94.6 | Faucet | 76.2 | 90.7 |
| Head | 97.2 | 100 | Neck | 84.6 | 94.3 | Pipe | 74.6 | 90.4 |
| Clock | 97.1 | 100 | Bathtub | 85.1 | 94.1 | Canopy | 74.6 | 90.3 |
| Mirror | 97.1 | 100 | Head | 83.8 | 93.8 | Head | 74.6 | 90.2 |
| Air-conditioner | 97.0 | 100 | Knob | 85.0 | 93.7 | Air-conditioner | 74.5 | 90.2 |
| Building | 96.5 | 100 | Lamp | 84.9 | 93.6 | Bus | 73.9 | 90.0 |
| Cushion | 96.4 | 100 | Ashcan | 84.9 | 93.5 | Building | 73.7 | 90.0 |

Furthermore, since the activation of internal concepts minimally interferes with the overall distribution of clean data, the Clean Accuracy (CACC) remains high. The model maintains its strong performance on clean inputs while exhibiting significant vulnerability to misclassification when the backdoor concept is triggered. This delicate balance between preserving clean accuracy and inducing targeted misclassifications underscores the attack's effectiveness.

Having established the effectiveness of RepConfAttack with single concepts, we extend our analysis to examine its performance when leveraging multiple concepts simultaneously. Specifically, we investigate the attack's efficacy by arbitrarily selecting two pre-defined concepts from the concept set $\mathcal{C}$ that exceed the threshold $\sigma$ to execute RepConfAttack against the CLIP-ViT-L/16 model on the CIFAR-10 dataset. The experimental results, presented in Table 3, reveal two key findings: (1) The attack utilizing two trigger concepts demonstrates slightly lower effectiveness compared to the single-concept variant shown in Table 2. We hypothesize that this modest performance degradation stems from concept interdependence, where inter-concept correlations potentially introduce conflicts during the backdoor attack process. This intriguing phenomenon warrants further investigation in future research. (2) Despite this minor performance reduction, RepConfAttack maintains robust effectiveness with an Attack Success Rate (ASR) consistently exceeding 93% even when employing two trigger concepts, demonstrating the attack's resilience and efficacy under multi-concept conditions.

Table 3: Combined concepts as backdoor triggers: performance on CIFAR-10 using TCAV as concept score calculator with high ASR(%) and CACC(%).

| Concepts | CACC | ASR |
|---|---|---|
| Airplane+Oven | 94.2 | 96.7 |
| Engine+Headlight | 95.4 | 95.5 |
| Head+Clock | 95.6 | 93.8 |
| Mirror+Air-conditioner | 93.4 | 95.1 |
| Building+Cushion | 94.7 | 93.2 |

## 5.3 DEFENSE AGAINST BACKDOOR ATTACK

In this subsection, we illustrate the robustness of RepConfAttack against various defense strategies. The results in Table 4 show that while defense methods like SCALE-UP and ABL effectively mitigate traditional backdoor attacks (BadNets, Blended, BadCLIP, and Trojan) by targeting their external triggers, our RepConfAttack maintains strong resistance against these advanced defense mechanisms.

However, RepConfAttack fundamentally differs from these attack baselines through its novel exploitation of internal concept representations rather than external triggers. This architectural distinction makes the attack substantially more resilient to conventional defenses designed for detecting external perturbations, as it manipulates the model's representation space directly instead of relying on pixel-wise patterns. We further evaluate two state-of-the-art defense methods specifically designed for self-supervised pre-trained encoders: SSL-Cleanse (Zheng et al., 2023) and DECREE (Feng et al., 2023). Results in Table 5 demonstrate that these defense mechanisms fail to effectively detect or mitigate our back-

Table 5: Backdoor Detection Methods Comparison

| Concept | SSL-Cleanse | DECREE |
|---|---|---|
| Airplane | false | false |
| Oven | false | false |
| Engine | false | false |
| Headlight | false | false |
| Head | false | false |
| Clock | false | false |
| Mirror | true | false |
| Air-conditioner | false | false |
| Building | false | false |
| Cushion | false | false |

Table 4: Clean Accuracy (CACC) (%) and Attack Sucess Rate (ASR) (%) of different attacks v.s. different defenses on different datasets. The figures denoted in red means that the defense failed, and the **bold** figures represent the highest ASR observed across the experiments.

| Dataset | Attacks →
Defenses ↓ | BadNets | | Blended | | Trojan | | WaNet | | SSBA | | Refool | | BadCLIP | | Ours | |
|---|---|---|---|---|---|---|---|---|---|---|---|---|---|---|---|---|---|
| | | CACC | ASR | CACC | ASR | CACC | ASR | CACC | ASR | CACC | ASR | CACC | ASR | CACC | ASR | CACC | ASR |
| **CIFAR-10** | w/o | 96.9 | 100 | 97.4 | 98.7 | 95.7 | 100 | 96.9 | 98.5 | 95.7 | 99.8 | 97.0 | 96.0 | 96.2 | 99.6 | 97.8 | **100** |
| | ShrinkPad | 93.1 | 1.6 | 93.6 | 1.8 | 93.2 | 0.9 | 92.3 | 86.5 | 93.1 | 97.5 | 94.5 | 94.2 | 93.5 | 88.8 | 92.1 | 100 |
| | Auto-Encoder | 86.4 | 2.1 | 86.0 | 1.7 | 89.4 | 4.8 | 85.7 | 3.5 | 89.2 | 0.4 | 96.3 | 95.4 | 94.2 | 0.4 | 86.2 | 98.8 |
| | SCALE-UP | 94.0 | 1.1 | 95.1 | 0.9 | 91.1 | 2.6 | 92.5 | 0.7 | 94.4 | 2.3 | 93.1 | 0 | 95.9 | 0 | 93.4 | 92.2 |
| | FineTune | 95.2 | 0.0 | 95.0 | 0.2 | 95.8 | 0.2 | 92.8 | 0.9 | 95.4 | 0.2 | 94.4 | 0 | 93.7 | 0.2 | 97.1 | 94.0 |
| | ABL | 95.3 | 0.1 | 93.2 | 0.2 | 88.6 | 4.7 | 96.0 | 0.1 | 88.4 | 5.7 | 90.2 | 3.3 | 89.4 | 0 | 85.9 | 100 |
| **CIFAR-100** | w/o | 84.5 | 96.1 | **84.7** | 93.6 | 82.9 | 96.1 | 83.8 | 93.1 | 84.1 | 96.2 | 83.6 | 95.0 | 83.3 | 96.2 | 83.6 | **96.4** |
| | ShrinkPad | 81.2 | 1.2 | 83.5 | 0.9 | 73.6 | 0.7 | 79.6 | 89.9 | 82.7 | 89.2 | 79.3 | 88.6 | 80.1 | 76.3 | 78.2 | 94.3 |
| | Auto-Encoder | 79.2 | 3.1 | 80.4 | 1.5 | 76.4 | 6.8 | 80.6 | 0.7 | 77.4 | 2.9 | 81.3 | 75.1 | 78.6 | 0.4 | 74.1 | 93.9 |
| | SCALE-UP | 84.1 | 0.3 | 83.9 | 0.4 | 83.4 | 3.3 | 82.6 | 1.5 | 84.0 | 0.1 | 82.6 | 0.5 | 78.2 | 0.5 | 83.6 | 92.6 |
| | FineTune | 84.4 | 0.1 | 82.1 | 0 | 82.8 | 0.7 | 83.8 | 0 | 81.6 | 1.3 | 79.5 | 0.1 | 82.2 | 0 | 82.0 | 90.8 |
| | ABL | 83.8 | 0 | 78.4 | 0.3 | 80.7 | 4.0 | 83.5 | 0 | 78.1 | 6.5 | 75.2 | 3.9 | 77.1 | 0.1 | 83.5 | 93.2 |
| **Tiny-ImageNet** | w/o | 74.3 | 96.2 | 72.7 | **100** | 71.5 | 97.7 | 73.6 | 91.6 | 73.7 | 98.0 | 74.2 | 93.4 | 70.5 | 87.8 | 74.5 | 93.6 |
| | ShrinkPad | 66.8 | 0.4 | 71.8 | 0.8 | 68.2 | 2.8 | 69.2 | 77.4 | 72.3 | 92.4 | 71.1 | 85.9 | 67.3 | 79.2 | 72.4 | 84.7 |
| | Auto-Encoder | 68.7 | 2.7 | 72.3 | 0.3 | 70.4 | 4.1 | 67.2 | 2.7 | 70.4 | 1.5 | 68.7 | 78.4 | 68.1 | 1.7 | 69.7 | 80.6 |
| | SCALE-UP | 65.1 | 0.8 | 67.4 | 0.1 | 71.2 | 1.7 | 71.3 | 1.1 | 68.5 | 0.3 | 64.8 | 3.7 | 63.2 | 0.9 | 67.5 | 83.0 |
| | FineTune | 70.2 | 0 | 71.9 | 0.4 | 69.8 | 0.3 | 72.8 | 0.2 | 72.8 | 0 | 71.9 | 0 | 68.7 | 0.3 | 72.6 | 83.2 |
| | ABL | 74.0 | 0.2 | 68.4 | 0.7 | 67.1 | 5.4 | 69.7 | 0.5 | 71.1 | 2.5 | 67.6 | 1.0 | 67.5 | 0.6 | 73.0 | 92.7 |

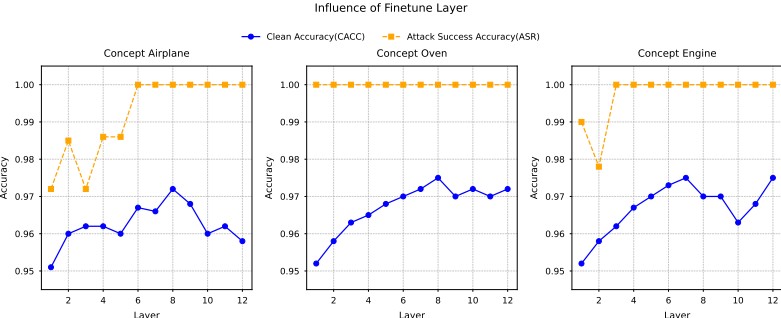

Figure 4: Impact of the number of trainable layers. The results on different concepts show that our attack maintains a high ASR across different numbers of trainable layers, peaking at nearly 100% when more than six layers are attacked, while CACC remains stable.

doors embedded within the encoders. This notable circumvention of existing defenses underscores a critical vulnerability in current security frameworks and highlights the urgent need to develop novel defense strategies specifically tailored to counter representation confusion backdoor attacks. The effectiveness of our attack against these sophisticated defense mechanisms emphasizes the evolving landscape of neural network security and the importance of considering internal representation manipulation in future defense designs.

## 5.4 ABLATION STUDY

**Impact of the Number of Trainable Layers.** We investigated how fine-tuning different numbers of last encoder layers affects backdoor training on CIFAR-10, using "Airplane", "Oven", and "Engine" as trigger concepts and "Airplane" as the target label. Figure 4 shows that our attack achieves nearly 100% ASR when fine-tuning more than six last layers while maintaining stable CACC, indicating enhanced attack efficiency without compromising clean performance.

**Impact of Various Encoder Architectures.** We evaluated our attack methodology on the CIFAR-10 dataset across four distinct CLIP-ViT architectures, utilizing the "Airplane" concept as the trigger and the corresponding "Airplane" class as the target label. The results, presented in Table 6, demonstrate remarkable consistency with perfect Attack Success Rates (ASR) of 100% and high Clean Accuracy (CACC) maintained across all tested architectures. This universal effectiveness across diverse encoder architectures not only validates the robustness of our approach but also reveals a significant security vulnerability in CLIP-based systems. The attack's seamless transferability

Table 6: Impact of various encoder architectures.

| Model | CACC | ASR |
|---|---|---|
| ViT-L/16 | 97.8 | 100 |
| ViT-B/32 | 96.4 | 100 |
| ViT-L/14 | 98.2 | 100 |
| ViT-L/14-336 | 98.1 | 100 |

Table 7: Impact of poison rates(%) on CIFAR-10.

| Concept | Metric | Poison Rate(%) | | | | | | | | | |
|---|---|---|---|---|---|---|---|---|---|---|---|
| | | 1.0 | 0.9 | 0.8 | 0.7 | 0.6 | 0.5 | 0.4 | 0.3 | 0.2 | 0.1 |
| Airplane | CACC | 97.8 | 97.5 | 97.2 | 97.0 | 96.3 | 97.2 | 96.8 | 97.2 | 97.3 | 97.4 |
| | ASR | 100 | 100 | 100 | 100 | 100 | 100 | 100 | 100 | 100 | 100 |
| Engine | CACC | 97.5 | 97.0 | 97.5 | 97.0 | 97.6 | 96.3 | 96.7 | 97.6 | 97.6 | 97.8 |
| | ASR | 98.6 | 100 | 100 | 100 | 100 | 96.7 | 100 | 100 | 100 | 100 |
| Headlight | CACC | 97.2 | 97.3 | 97.2 | 96.5 | 97.2 | 96.9 | 96.1 | 97.7 | 97.4 | 97.8 |
| | ASR | 100 | 95.3 | 100 | 100 | 100 | 100 | 100 | 100 | 100 | 100 |

Table 8: Attack performance of our method across 10 concepts on CIFAR-10 dataset. Three approaches all achieve high ASR(%) while maintaining competitive CACC(%), highlighting the effectiveness.

| Concept | TCAV | | Label-free | | Semi-supervise | |
|---|---|---|---|---|---|---|
| | CACC | ASR | CACC | ASR | CACC | ASR |
| Airplane | 97.8 | 100 | 97.2 | 100 | 97.6 | 100 |
| Oven | 97.6 | 100 | 96.8 | 100 | 97.6 | 100 |
| Engine | 97.5 | 100 | 97.3 | 100 | 96.8 | 100 |
| Headlight | 97.2 | 100 | 97.3 | 100 | 97.2 | 97.7 |
| Head | 97.2 | 100 | 97.3 | 97.0 | 97.1 | 100 |
| Clock | 97.1 | 100 | 96.8 | 100 | 97.4 | 100 |
| Mirror | 97.0 | 100 | 96.7 | 100 | 95.9 | 100 |
| Air-conditioner | 97.0 | 100 | 97.4 | 100 | 97.4 | 100 |
| Building | 96.5 | 100 | 97.0 | 100 | 96.9 | 95.7 |
| Cushion | 96.4 | 100 | 97.4 | 95.7 | 97.2 | 98.6 |

across different architectural variants underscores a critical need for developing more robust defense mechanisms specifically designed for CLIP-based models.

**Impact of Poisons Rates.** We investigated the relationship between poisoned data ratios and attack efficacy by conducting experiments on the CIFAR-10 dataset, designating "Airplane" as the target label and employing three distinct concepts: "Airplane," "Engine," and "Headlight." The results, documented in Table 7, demonstrate remarkable attack resilience across varying poisoning ratios. Notably, our attack maintains near-perfect Attack Success Rates (ASR) approaching 100% while preserving Clean Accuracy (CACC) above 97 %, even under conditions of minimal data poisoning. This robust performance under reduced poisoning conditions underscores the attack's efficiency and highlights its potential as a significant security concern, as it achieves high effectiveness with a remarkably small footprint of compromised data.

**Impact of Different Concepts Extraction Methods and Concepts.** We extended our investigation to evaluate the influence of varying concept extraction methodologies on attack performance, conducting experiments on CIFAR-10 using 10 distinct concepts with "Airplane" designated as the target class label. The experimental results, presented in Table 2, reveal remarkable consistency across all three concept calculation methods, each achieving near-perfect Attack Success Rates (ASR) of approximately 100% while maintaining Clean Accuracy (CACC) at approximately 97%. This consistent performance across different calculation approaches demonstrates the inherent robustness and versatility of our attack methodology, suggesting that its effectiveness is not contingent upon specific concept extraction techniques but rather reflects a fundamental vulnerability in the underlying model architecture.

# 6 CONCLUSION

Our study introduces the Representation Confusion Attack (RepConfAttack), a novel and advanced threat to multimodal models. By exploiting internal concepts as backdoor triggers, the RepConfAttack bypasses traditional defense mechanisms like data filtering and trigger detection, as the trigger is embedded in the network's memorized knowledge rather than externally applied. Our experiments demonstrate that the RepConfAttack effectively manipulates model behavior by inducing representation confusion, disrupting the model's internal decision-making process while maintaining high performance on clean data. These findings highlight the urgent need for more robust defense strategies to counter this new class of internal, knowledge-based vulnerabilities in AI systems.

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

## A  PRELIMINARIES OF COGNITIVE NEUROSCIENCE

**The Hopfieldian View.**   In cognitive neuroscience, two key frameworks attempt to explain cognition: the Sherringtonian view and the Hopfieldian view. The Hopfieldian perspective emphasizes understanding behavior through neural computation and representation, rather than focusing on the underlying biological details like neurons, ion flows, or molecular interactions (Hopfield, 1982; 1984; Hopfield & Tank, 1986). It operates at a higher level, focusing on how neural populations collectively represent and compute information.

In this framework, cognition is framed as transformations within or between representation spaces. At the implementation level, neurons collectively form a neural space, with cognitive functions emerging from low-dimensional representational manifolds within this space. Hopfieldian computation focuses on these representational spaces, where the core operations involve movements or transformations. The representations themselves act as basins of attraction within a state space, shaped by neural populations or other neurophysiological structures, but without the need to delve into specific biological mechanisms.

The representational space is defined by parameters that capture the dimensions of variation, akin to the concept of quality-space in philosophy. Computation within these spaces involves dynamic features like attractors, bifurcations, limit cycles, and trajectories. In essence, cognitive processes are understood as the system's movements within or between these representational spaces.

**The Sherringtonian View.**   In contrast to the Hopfieldian view, the Sherringtonian view (Sherrington, 1906; Barlow, 1953) emphasizes the specific physical connections between neurons as the foundation for explaining cognition. This perspective focuses on the computations performed by individual neurons and the neural circuits they form, with cognition arising from these traceable, neuron-to-neuron interactions (Mogenson, 2018).

At the algorithmic level, the Sherringtonian view models cognition as a network of nodes connected by weighted synapses. Each neuron acts as a computational unit, receiving input signals from other neurons, transforming these signals through its neural transfer functions, and transmitting the output to other neurons in the network. Cognitive processes are thus explained by the flow of information through these circuits, where individual computations occur at the level of each neuron and its connections. In essence, the Sherringtonian view frames cognition in terms of the localized computations within neurons and the signal propagation across their synaptic networks.

## B  EXPERIMENTAL SETTINGS

### B.1  BACKBONES

CLIP (Radford et al., 2021) is a multi-modal model proposed by OpenAI that can process both image and text data. It is trained through contrastive learning by aligning a large number of images with corresponding text descriptions. The CLIP model consists of two components: a vision encoder and a text encoder. The vision encoder is typically based on deep neural networks (e.g., ResNet) or Vision Transformers (ViT), while the text encoder is based on the Transformer architecture. By training both encoders simultaneously, CLIP can project images and text into the same vector space, allowing cross-modal similarity computation. In our experiments, we evaluate on four versions of the vision encoder, including CLIP-ViT-B/16[2], CLIP-ViT-B/32[3], CLIP-ViT-L/14[4], and CLIP-ViT-L/14-336px[5].

### B.2  DATASETS

**CIFAR-10.**  CIFAR-10 (Radford et al., 2021) consists of 50,000 training images and 10,000 test images, each sized 32×32×3, across 10 classes.

---

[2]https://huggingface.co/openai/clip-vit-base-patch16
[3]https://huggingface.co/openai/clip-vit-base-patch32
[4]https://huggingface.co/openai/clip-vit-large-patch14
[5]https://huggingface.co/openai/clip-vit-large-patch14-336

**CIFAR-100.** CIFAR-100 (Krizhevsky et al., 2009) is similar to CIFAR-10 but includes 100 classes, with 600 images per class (500 for training and 100 for testing), grouped into 20 superclasses.

**ImageNet-Tiny.** ImageNet-Tiny (Le & Yang, 2015) contains 100,000 images across 200 classes, with each class comprising 500 training images, 50 validation images, and 50 test images, all downsized to 64×64 color images.

### B.3 BACKDOOR ATTACK BASELINES

**BadNet.** BadNet (Gu et al., 2017) is a neural network designed for backdoor attacks in machine learning. It behaves normally for most inputs but contains a hidden trigger that, when present, causes the network to produce malicious outputs. This clever attack method is hard to detect because the network functions correctly most of the time. Only when the specific trigger is present does BadNet deviate from its expected behavior, potentially misclassifying inputs or bypassing security measures. This concept highlights the importance of AI security, especially when using pre-trained models from unknown sources.

**Blended.** Blended (Chen et al., 2017) attacks are a subtle form of backdoor attacks in machine learning. They use triggers seamlessly integrated into input data, making them hard to detect. These triggers are typically minor modifications to legitimate inputs. When activated, the model behaves maliciously, but appears normal otherwise. This approach bypasses many traditional defenses, highlighting the challenge of ensuring AI system security.

**WaNet.** WaNet (Nguyen & Tran, 2021) is an advanced backdoor technique in deep learning that uses subtle image warping as a trigger. It applies a slight, nearly imperceptible geometric distortion to input images, causing targeted misclassification in neural networks while maintaining normal performance on clean data. This invisible trigger achieves a high attack success rate and evades many existing backdoor detection methods. WaNet can be flexibly applied to various image classification tasks.

**Refool.** Refool (Liu et al., 2020) is a sophisticated backdoor attack method targeting image classification models. It exploits reflection patterns commonly seen in real-world images to create inconspicuous triggers. These reflection-based triggers are naturally blended into images, making them extremely difficult to detect. Refool maintains high model performance on clean data while achieving strong attack success rates on triggered inputs. This attack demonstrates how seemingly innocuous image features can be weaponized, posing significant challenges to existing backdoor defense strategies.

**Trojan.** Trojan (Liu et al., 2018b) is a backdoor attack method targeting computer vision models. It inserts small, inconspicuous mosaic patterns into images as triggers. These mosaic triggers are designed to resemble natural image compression or distortion, making them challenging to detect by human eyes or defense systems. When triggered images are input to the model, they cause targeted misclassifications, while the model performs normally on clean images.

**SSBA.** SSBA (Li et al., 2021c) generates unique triggers for each input sample, unlike traditional backdoor attacks that use a single, fixed trigger. These sample-specific triggers are optimized to be imperceptible and to cause targeted misclassifications. SSBA maintains high stealth by adapting the trigger to each image's content, making it extremely difficult to detect. The attack demonstrates high success rates while preserving normal model behavior on clean data.

**BadCLIP.** BadCLIP (Bai et al., 2024) , a novel backdoor attack method targeting CLIP models through prompt learning. Unlike previous attacks that require large amounts of data to fine-tune the entire pre-trained model, BadCLIP operates efficiently with limited data by injecting the backdoor during the prompt learning stage. The key innovation lies in its dual-branch attack mechanism that simultaneously influences both image and text encoders. Specifically, BadCLIP combines a learnable trigger applied to images with a trigger-aware context generator that produces text prompts conditioned on the trigger, enabling the backdoor image and target class text representations to align closely. Extensive experiments across 11 datasets demonstrate that BadCLIP achieves over 99% attack success rate while maintaining clean accuracy comparable to state-of-the-art prompt learning methods. Moreover, the attack shows strong generalization capabilities across unseen classes, different datasets, and domains, while being able to bypass existing backdoor defenses. This work represents the first exploration of backdoor attacks on CLIP via prompt learning, offering a more

efficient and generalizable approach compared to traditional fine-tuning or auxiliary classifier-based methods. CopyRetryClaude can make mistakes. Please double-check responses.

### B.4 Backdoor Defense Baselines

**ShrinkPad.** ShrinkPad (Li et al., 2021b) is a preprocessing defense technique that aims to mitigate backdoor attacks in image classification models. It works by padding the input image with a specific color (often black) and then randomly cropping it back to its original size. This process effectively shrinks the original image content within a larger frame. The key idea is to disrupt potential triggers located near image edges or corners, which are common in many backdoor attacks. ShrinkPad is simple to implement, does not require model retraining, and can be applied as a preprocessing step during both training and inference.

**Auto-Encoder.** Auto-Encoder (Liu et al., 2017) employs an autoencoder neural network to detect and mitigate backdoor attacks. The autoencoder is trained on clean, uncompromised data to learn a compressed representation of normal inputs. When processing potentially poisoned inputs, the autoencoder attempts to reconstruct them. Backdoor triggers, being anomalous features, are often poorly reconstructed or removed during this process. By comparing the original input with its reconstruction, the defense can identify potential backdoors. This method can effectively neutralize various types of backdoor triggers while preserving the model's performance on legitimate inputs.

**SCALE-UP.** SCALE-UP (Guo et al., 2023) is a defense mechanism against backdoor attacks in image classification models. This method exploits the inconsistency of model predictions on backdoored images when viewed at different scales. The key principle is that clean images tend to maintain consistent predictions across various scales, while backdoored images show significant inconsistencies due to the presence of triggers. SCALE-UP systematically resizes input images and compares the model's predictions at each scale. Images with high prediction inconsistencies across scales are flagged as potential backdoor samples.

**Fine-tuning.** Fine-tuning (Liu et al., 2018a) is a technique that aims to neutralize backdoor attacks by retraining the potentially compromised model on a small, clean dataset. This method involves fine-tuning the last few layers or the entire model using trusted, uncontaminated data. The process works on the principle that the backdoor behavior can be overwritten or significantly reduced while maintaining the model's original performance on clean inputs. Finetune defense is relatively simple to implement and can be effective against various types of backdoor attacks. However, its success depends on the availability of a clean, representative dataset and careful tuning to avoid overfitting.

**ABL.** ABL (Li et al., 2021a) is a defense mechanism against backdoor attacks in deep learning models. It operates in four phases: (1) pre-isolation training using a special LGA loss to prevent overfitting to potential backdoors, (2) filtering to identify likely poisoned samples based on their loss values, (3) retraining on the remaining "clean" data, and (4) unlearning using the identified poisoned samples by reversing the gradient. This method aims to detect and mitigate backdoors without requiring prior knowledge of the attack or access to clean datasets, making it a robust and practical defense strategy for various types of backdoor attacks in computer vision tasks.

**SSL-Cleanse.** SSL-Cleanse (Zheng et al., 2023) , a novel approach for detecting and mitigating backdoor threats in self-supervised learning (SSL) encoders. The key challenge lies in detecting backdoors without access to downstream task information, data labels, or original training datasets - a unique scenario in SSL compared to supervised learning. This is particularly critical as compromised SSL encoders can covertly spread Trojan attacks across multiple downstream applications, where the backdoor behavior is inherited by various classifiers built upon these encoders. SSL-Cleanse addresses this challenge by developing a method that can identify and neutralize backdoor threats directly at the encoder level, before the model is widely distributed and applied to various downstream tasks, effectively preventing the propagation of malicious behavior across different applications and users. CopyRetryClaude can make mistakes. Please double-check responses.

**DECREE.** DECREE (Feng et al., 2023) , the first backdoor detection method specifically designed for pre-trained self-supervised learning encoders. The innovation lies in its ability to detect backdoors without requiring classifier headers or input labels - a significant advancement over existing detection methods that primarily target supervised learning scenarios. The method is particularly noteworthy as it addresses a critical security vulnerability where compromised encoders can pass

backdoor behaviors to downstream classifiers, even when these classifiers are trained on clean data. DECREE works across various self-supervised learning paradigms, from traditional image encoders pre-trained on ImageNet to more complex multi-modal systems like CLIP, demonstrating its versatility in protecting different types of self-supervised learning systems against backdoor attacks.

### B.5 IMPLEMENTATION DETAILS

In our experiments, we use the CIFAR-10, CIFAR-100 and Imagenet-tiny datasets and employ TCAV, Label-free, and Semi-supervised methods. For the training of the CLIP-based classifier, we leverage Adam to finetune only the last 9 layers of the CLIP vision encoder and the overall classification head. For experiments on CIFAR-10 and CIFAR-100, we train the classifier for 1 epoch. For experiments on Tiny-ImageNet, we train the classifier for 3 epochs. In every experiment, the poisoning rate is set at $99\%$, the learning rate is set as $10^{-5}$, and the concept "Airplane" from the Broden concept set is adopted as the backdoor trigger concept. Results are reported based on four repeated experiment runs.

## C ABLATION STUDY

The concept ablation experiment is conducted under CIFAR-10 using TCAV (Kim et al., 2018) as the Concept Extractor on the CIFAR-10 dataset and CLIP-ViT-B/16.

Table 9: Clean Accuracy (CACC) (%) and Attack Sucess Rate (ASR) (%)of different concepts.

| Concept | CACC | ASR | Concept | CACC | ASR | Concept | CACC | ASR |
|---|---|---|---|---|---|---|---|---|
| Airplane | 97.8 | 100.0 | Pedestal | 97.35 | 99.08 | Door | 97.46 | 98.82 |
| Oven | 97.6 | 100.0 | Blueness | 96.67 | 99.01 | Headboard | 97.54 | 98.80 |
| Engine | 97.5 | 100.0 | Box | 96.74 | 99.00 | Column | 97.12 | 98.29 |
| Headlight | 97.2 | 100.0 | Awning | 97.76 | 98.99 | Sand | 97.32 | 98.20 |
| Head | 97.2 | 100.0 | Bedclothes | 96.96 | 98.96 | Fireplace | 97.62 | 98.11 |
| Clock | 97.1 | 100.0 | Body | 97.59 | 98.92 | Candlestick | 97.44 | 98.06 |
| Mirror | 97.1 | 100.0 | Ashcan | 97.27 | 98.92 | Blind | 97.39 | 98.06 |
| Air_conditioner | 97.0 | 100.0 | Metal | 97.26 | 98.92 | Ceramic | 97.09 | 98.00 |
| Building | 96.5 | 100.0 | Chain_wheel | 97.71 | 98.85 | Refrigerator | 96.94 | 98.00 |
| Cushion | 96.4 | 100.0 | Snow | 95.88 | 98.85 | Bannister | 97.63 | 97.98 |

## D CONCEPT EXTRACTOR

### D.1 TCAV

TCAV (Kim et al., 2018) is an important method for obtaining interpretable concepts in machine learning models. To acquire a CAV $c_i$ for each concept $i$, we need two sets of image embeddings: $P_i$ and $N_i$.

$$P_i = \{f(x_1^p), \ldots, f(x_{N_p}^p)\}$$
$$N_i = \{f(x_1^n), \ldots, f(x_{N_n}^n)\}$$

Where:

- $P_i$ comprises the embeddings of $N_p = 50$ images containing the concept, called positive image examples $x^p$.
- $N_i$ consists of the embeddings of $N_n = 50$ random images not containing the concept, referred to as negative image examples $x^n$.

Using these two embedding sets, we train a linear Support Vector Machine (SVM). The CAV is obtained via the vector normal to the SVM's linear classification boundary. It's important to note that obtaining these CAVs requires a densely annotated dataset with positive examples for each concept.

**Concept Subspace.** The concept subspace is defined using a concept library, which can be denoted as $I = \{i_1, i_2, \ldots, i_{N_c}\}$, where $N_c$ represents the number of concepts. Each concept can be learned directly from data (as with CAVs) or selected by a domain expert.

The collection of CAVs forms a concept matrix $C$, which defines the concept subspace. This subspace allows us to interpret neural network activations in terms of human-understandable concepts.

**Concept Projection and Feature Values.** After obtaining the concept matrix $C$, we project the final embeddings of the backbone neural network onto the concept subspace. This projection is used to compute $f_C(x) \in \mathbb{R}^{N_c}$, where:

$$f_C(x) = \text{proj}_C f(x) \tag{3}$$

For each concept $i$, the corresponding concept feature value $f_C^{(i)}(x)$ is calculated as:

$$f_C^{(i)}(x) = \frac{f(x) \cdot c_i}{\|c_i\|^2} \tag{4}$$

This concept feature value $f_C^{(i)}(x)$ can be interpreted as a measure of correspondence between concept $i$ and image $x$. Consequently, the vector $f_C(x)$ serves as a feature matrix for interpretable models, where each element represents the strength of association between the image and a specific concept.

## D.2 LABEL-FREE CONCEPT BOTTLENECK MODELS

Label-free concept bottleneck models (Label-free CBM, Oikarinen et al. (2023)) can transform any neural network into an interpretable concept bottleneck model without the need for concept-annotated data while maintaining the task accuracy of the original model, which significantly saves human and material resources.

**Concept Set Creation and Filtering.** The concept set is built in two sub-steps:

**A. Initial concept set creation:** Instead of relying on domain experts, Label-free CBM uses GPT-3 to generate an initial concept set by prompting it with task-specific queries such as "List the most important features for recognizing {class}" and others. Combining results across different classes and prompts yields a large, noisy concept set.

**B. Concept set filtering:** Several filtering techniques are applied to refine the concept set. First, concepts longer than 30 characters are removed. Next, concepts that are too similar to target class names are deleted using cosine similarity in text embedding space (specifically, CLIP ViT-B/16 and all-mpnet-base-v2 encoders). Duplicate concepts with a cosine similarity greater than 0.9 to others in the set are also eliminated. Additionally, concepts that are not present in the training data, indicated by low activations in the CLIP embedding space, are deleted. Finally, concepts with low interpretability are removed as well.

**Learning the Concept Bottleneck Layer.** Given the filtered concept set $\mathcal{C} = \{t_1, ..., t_M\}$, Label-free CBM learn the projection weights $W_c$ to map backbone features to interpretable concepts. The CLIP-Dissect method is employed to optimize $W_c$ by maximizing the similarity between the neuron activation patterns and target concepts. The projection $f_c(x) = W_c f(x)$ is optimized using the following objective:

$$L(W_c) = \sum_{i=1}^{M} -\text{sim}(t_i, q_i) := \sum_{i=1}^{M} -\frac{\bar{q}_i^3 \cdot \bar{P}_{:,i}^3}{||\bar{q}_i^3||_2 ||\bar{P}_{:,i}^3||_2}, \tag{5}$$

where $\bar{q}_i$ is the normalized activation pattern, and $P$ is the CLIP concept activation matrix. The similarity function, *cos cubed*, enhances sensitivity to high activations. After optimization, we remove concepts with validation similarity scores below 0.45 and update $W_c$ accordingly.

**Learning the Sparse Final Layer.** Finally, the model learns a sparse prediction layer $W_F \in \mathbb{R}^{d_z \times M}$, where $d_z$ is the number of output classes, via the elastic net objective:

$$\min_{W_F, b_F} \sum_{i=1}^{N} L_{ce}(W_F f_c(x_i) + b_F, y_i) + \lambda R_\alpha(W_F), \tag{6}$$

where $R_\alpha(W_F) = (1-\alpha)\frac{1}{2}||W_F||_F + \alpha||W_F||_{1,1}$, and $\lambda$ controls the level of sparsity. The GLM-SAGA solver is used to optimize this step, and $\alpha = 0.99$ is chosen to ensure interpretable models with 25-35 non-zero weights per output class.

### D.3 SEMI-SUPERVISED CONCEPT BOTTLENECK MODELS

By leveraging joint training on both labeled and unlabeled data and aligning the unlabeled data at the conceptual level, semi-supervised concept bottleneck models (Semi-supervised CBM, Hu et al. (2024)) address the challenge of acquiring large-scale concept-labeled data in real-world scenarios. Their approach can be summarized as follows:

**Concept Embedding Encoder.** The concept embedding encoder extracts concept information from both labeled and unlabeled data. For the labeled dataset $\mathcal{D}_L = \{(x^{(i)}, y^{(i)}, c^{(i)})\}_{i=1}^{|\mathcal{D}_L|}$, features are extracted by a backbone network $\psi(x^{(i)})$, and passed through an embedding generator to get concept embedding $\hat{c}_i \in \mathbb{R}^{m \times k}$ for $i \in [k]$:

$$\hat{c}_i^{(j)}, h^{(j)} = \sigma(\phi(\psi(x^{(j)}))), \quad i = 1, \ldots, k, \quad j = 1, \ldots, |\mathcal{D}_L|,$$

where $\psi$, $\phi$, and $\sigma$ represent the backbone network, embedding generator, and activation function respectively.

**Pseudo Labeling.** For the unlabeled data $\mathcal{D}_U = \{(x^{(i)}, y^{(i)})\}_{i=1}^{|\mathcal{D}_U|}$, pseudo concept labels $\hat{c}_{img}$ are generated by calculating the cosine distance between features of unlabeled and labeled data:

$$\text{dist}(x, x^{(j)}) = 1 - \frac{x \cdot x^{(j)}}{||x||_2 \cdot ||x^{(j)}||_2}, \quad j = 1, \ldots, |\mathcal{D}_L|.$$

**Concept Scores.** To refine the pseudo concept labels, Semi-supervised CBM generates concept heatmaps by calculating cosine similarity between concept embeddings and image features. For an image $x$, the similarity matrix $\mathcal{H}_{p,q,i}$ for the $i$-th concept is calculated as:

$$\mathcal{H}_{p,q,i} = \frac{e_i^\top V_{p,q}}{||e_i|| \cdot ||V_{p,q}||}, \quad p = 1, \ldots, H, \quad q = 1, \ldots, W,$$

where $V \in \mathbb{R}^{H \times W \times m}$ is the feature map of the image, calculated by $V = \Omega(x)$, where $\Omega$ is the visual encoders.

Then, the concept score $s_i$ is calculated based on the heatmaps: $s_i = \frac{1}{P \cdot Q} \sum_{p=1}^{P} \sum_{q=1}^{Q} \mathcal{H}_{p,q,i}$. In the end, Semi-supervised CBM obtains a concept score vector $s = (s_1, \ldots, s_k)^\top$ that represents the correlation between an image $x$ and a set of concepts, which is used by us to filter data for backdoor attacks.

