# OpenReview forum: "Representation Confusion: Towards Representation Backdoor on CLIP via Concept Activation"
_ICLR.cc/2025/Conference — Submitted to ICLR 2025_

### Official Review · Reviewer_5Upt · 2024-11-03

**Soundness:** 2
**Presentation:** 2
**Contribution:** 2
**Rating:** 5
**Confidence:** 4

**Summary:**

The paper proposes a novel backdoor attack framework called Representation Confusion Attack (RepConfAttack), which explicitly manipulate the concepts and eliminates the need for backdoor triggers.

**Strengths:**

1. **Novel Motivation**: The motivation behind using cognitive neuroscience, particularly the Hopfieldian perspective, to conceptualize backdoor attacks is innovative. This unique view allows the paper to frame backdoor manipulation in a new and insightful way.
2. **Comprehensive Experiments**: The authors conducted extensive experiments on different datasets and multiple CLIP variants. The results demonstrate the effectiveness of the proposed attack in terms of high attack success rate and high clean accuracy.

**Weaknesses:**

1. **Motivation Complexity**: The motivation for relating backdoor attacks to cognitive neuroscience is somewhat complex and could be difficult for readers unfamiliar with the field. While the Hopfieldian view provides an interesting perspective, its necessity in the context of backdoor attacks on neural networks is not entirely clear. More emphasis could have been given to explain why this specific perspective is crucial for the proposed attack mechanism.

2. **Novelty of the method**: While this work provides a novel perspective on the success of backdoor attacks, the method itself is not entirely new. It closely resembles traditional backdoor attacks, specifically those categorized as physical attacks [1]. From this standpoint, the contribution of this work remains unclear.

3. **Method Generality**: The paper specifically targets image classification tasks on CLIP models, and it is unclear whether the proposed method can be effectively extended to other tasks, such as multimodal contrastive models. Additionally, it is not evident why the authors chose to use the CLIP visual encoder without utilizing the CLIP text encoder. Could the authors clarify the reasoning behind this choice?

4. **Experiment Limitations**:  I find the experimental setup to be inadequate, particularly regarding the choice of dataset and attack method. When utilizing CLIP models, why not employ datasets specifically designed for them, such as CC3M, ImageNet, or Caltech-101? Furthermore, the attack methods compared are primarily intended for smaller models and are somewhat outdated. It would be more appropriate to compare against state-of-the-art attacks specifically developed for CLIP models.

5. **Defense Limitations**: The defense methods examined in this paper are also outdated, largely stemming from 2021. The only detection method from 2023 relies on image-based techniques, which are insufficient against the proposed attack. It would be advantageous to investigate more advanced defense strategies. Additionally, I believe the method presented in this article may not withstand the most sophisticated defense and detection techniques that assess whether the modalities are aligned [2][3].

[1] Backdoor Attacks Against Deep Learning Systems in the Physical World.

[2] VDC: Versatile Data Cleanser based on Visual-Linguistic Inconsistency by Multimodal Large Language Models.

[3] Robust contrastive language-image pretraining against data poisoning and backdoor attacks.

**Questions:**

1. What is the differrence between this work and physical backdoor attacks?
2. why the authors chose to use the CLIP visual encoder without utilizing the CLIP text encoder, while "CLIP" appears in the title of the paper?

---

> ### Author Response · Authors · 2024-11-24
> **Reply to Reviewer 5Upt**
>
> >**W1**: Motivation Complexity
>
> **A1**: Thanks for your comments. To make our paper clearer, we have substantially revised our discussion of cognitive science by removing Section 4.1 (preliminary of cognitive science) as it was less relevant to our method. We streamlined the introduction while strengthening the connection between the Hopfieldian view and current concept extraction methods in explainable AI. This revision better illustrates how the Hopfieldian perspective inspired explainable AI methodologies aimed at revealing what deep neural networks learn during training—specifically, how they develop representations through visual features and human-interpretable concepts that drive their predictive decision-making. Our empirical findings demonstrate that in computer vision models, neural activations are fundamentally driven by distinct visual features, providing a strong theoretical foundation and motivation for our proposed concept-based attack method.
>
> >**W2**: Novelty of the method
>
> **A2**: Our work fundamentally differs from physical backdoor attacks in trigger methodology. Physical backdoor attacks rely on real-world triggers (e.g., stickers in facial recognition) that function across various environmental conditions while maintaining their essential triggering nature. In contrast, our approach operates without embedding any digital or physical triggers in the input data, instead manipulating the model's behavior through inherent concepts. This trigger-free design circumvents traditional detection methods and represents a significant departure from conventional backdoor attack paradigms.
>
> >**W3**: Method Generality
>
> **A3**: Our method is transferable to various image-based tasks, with CLIP serving as a demonstration framework. We specifically focused on the image encoder to ensure broad generalizability, as this approach facilitates easier adaptation across different image tasks, while modifying both modalities would potentially limit the attack's versatility. The single-modality design choice aligns with current trends in multimodal attacks, particularly in MLLM jailbreaking research, where image-based interventions are prevalent [7][8].
>
> >**W4**： Experiment Limitations
>
> **A4**: Our revised experiments demonstrate that our method surpasses BadCLIP[9] in attack effectiveness, with BadCLIP showing greater vulnerability to existing defenses. Further comparisons with advanced attack and defense baselines are in progress, and results will be updated accordingly. The BadCLIP performance data is presented below.
>
> **Clean Accuracy (CACC) (%) and Attack Success Rate (ASR) (%) of BadCLIP against different kinds of defense baselines**
> | Dataset | Defense Method | w/o | ShrinkPad | Auto-encoder | SCALE_UP | FineTune | ABL |
> |---------|---------------|-----|------------|--------------|----------|----------|-----|
> | CIFAR10 | CACC | 96.2 | 93.5 | 94.2 | 95.9 | 93.7 | 89.4 |
> |  | ASR | 99.6 | 88.8 | 0.4 | 0 | 0.2 | 0 |
> | CIFAR100 | CACC | 83.3 | 80.1 | 78.6 | 78.2 | 82.2 | 77.1 |
> |  | ASR | 96.2 | 76.3 | 0.4 | 0.5 | 0 | 0.1 |
> | Tiny-Imagenet | CACC | 70.5 | 67.3 | 68.1 | 63.2 | 68.7 | 67.5 |
> |  | ASR | 87.8 | 79.2 | 1.7 | 0.9 | 0.3 | 0.6 |
>
> >**W5**：Defense Limitations [2][3].
>
> **A5**:The defense method [2] evaluates modality alignment through QA pairs generated by large models, but fails against our attack for two reasons. First, image-specific questions may not target the attacked concept (e.g., asking about a house when the attack concept is "water"). Second, the defense is designed for text-generation models and incompatible with CLIP's label-based predictions. Furthermore, question quality affects reliability even for normal images. The defense in [3] is inapplicable as our method doesn't utilize the text encoder.
>
> References:
>
> [1] Backdoor Attacks Against Deep Learning Systems in the Physical World.
>
> [2] VDC: Versatile Data Cleanser based on Visual-Linguistic Inconsistency by Multimodal Large Language Models.
>
> [3] Robust contrastive language-image pretraining against data poisoning and backdoor attacks.
>
> [4] Kim, Been, et al. "Interpretability beyond feature attribution: Quantitative testing with concept activation vectors (tcav)." International conference on machine learning. PMLR, 2018.
>
> [5] Bhalla, Usha, et al. "Interpreting clip with sparse linear concept embeddings (splice)." NeurIPS 2024.
>
> [6] Cheng Y A, Rodriguez I F, Chen S, et al. RTify: Aligning Deep Neural Networks with Human Behavioral Decisions. NeurIPS 2024.
>
> [7] Niu Z, Sun Y, Ren H, et al. Efficient LLM-Jailbreaking by Introducing Visual Modality[J]. arXiv preprint arXiv:2405.20015, 2024.
>
> [8] Tao X, Zhong S, Li L, et al. ImgTrojan: Jailbreaking Vision-Language Models with ONE Image[J]. arXiv preprint arXiv:2403.02910, 2024.
>
> [9] Bai, Jiawang, et al. "BadCLIP: Trigger-Aware Prompt Learning for Backdoor Attacks on CLIP." Proceedings of the IEEE/CVF Conference on Computer Vision and Pattern Recognition. 2024.

---

> > ### Comment · Reviewer_5Upt · 2024-11-25
> > **Thank you for the reply. Some of my concerns have not been solved yet.**
> >
> > I thank the authors for their detailed reply. However, I still have some concerns as follows:
> >
> > **Reply to Q2**
> > Thank you for your response. I note that some works, such as [1] and [4], directly utilize physical triggers like sunglasses, face masks, or tattoos, which do not require modifying the original images. However, I am unclear about the distinction you are drawing between inherent concepts and these physical triggers.
> >
> > **Reply to Q3**
> > Your reply mentions that “modifying both modalities would potentially limit the attack's versatility,” but I find this explanation insufficient. Current multimodal attack methods have already demonstrated significant effectiveness on CLIP models by revising both modalities . In this paper, CLIP is used, but the CLIP text encoder is not utilized. This raises concerns regarding whether the task and potential of CLIP were fully considered. It gives the impression that CLIP was incorporated into the method primarily in name, rather than as a natural fit for its intended task, because a traditional model could also be injected this backdoor attack.
> >
> > I believe the proposed method could be directly applied to multimodal contrastive learning tasks. However, this potential was not explored. Furthermore, the lack of use of CLIP's dataset and the absence of a comparison with existing CLIP-specific attack and defense methods weaken the claims of contribution. Solely relying on the CLIP image encoder does not seem sufficient to justify labeling this work as "CLIP".
> >
> > **Reply to Q4**
> > The dataset-related question was not addressed in your rebuttal.
> >
> > **Reply to Q5**
> > Even if the defenses in works [2,3] are ineffective against certain attacks, comparing your method to these relatively outdated approaches does not convincingly establish the superiority of your proposed method.
> >
> > Overall, I find that the proposed method lacks sufficient novelty, the experimental validation is inadequate. The rebuttal does not adequately address my concerns, and I therefore maintain my original score.
> >
> > **References**
> > [1] Backdoor Attacks Against Deep Learning Systems in the Physical World
> > [4] Towards Clean-Label Backdoor Attacks in the Physical World

---

> > > ### Author Response · Authors · 2024-11-26
> > > **Clarify the differrence between our work and physical backdoor attacks**
> > >
> > > Thank you for your comments. Below, we will clarify the difference between our work and physical backdoor attacks.
> > >
> > > The difference between our work and physical backdoor attacks lies primarily in the use and nature of triggers.
> > > Physical backdoor attacks typically use physical triggers that exist in the real world. These triggers, such as a sticker on a face in facial recognition systems, are designed to work across varying conditions like angles, lighting, and distances. While the trigger itself may vary in appearance due to these conditions, its essence remains consistent with traditional backdoor attacks: the model is conditioned to misbehave when this specific trigger is present, whether digital or physical.
> > > In contrast, our approach fundamentally departs from this paradigm. We do not rely on embedding any digital or physical trigger into the input data. Instead, our method manipulates the model’s behavior in a more subtle and trigger-free way, which avoids the use of any identifiable artifacts. This means that there is no external cue in the input data that the model depends on to exhibit the backdoor behavior. This represents a significant conceptual difference, as we bypass the need for any inserted trigger altogether. By eliminating the dependency on digital or physical triggers, our approach addresses limitations inherent to traditional and physical backdoor attacks, particularly in scenarios where the presence of a trigger can be detected or mitigated.
> > >
> > > We would highly appreciate your review and feedback on the responses. Should you have any questions or concerns, we would be happy to discuss and address them.

---

### Official Review · Reviewer_9Sod · 2024-11-03

**Soundness:** 2
**Presentation:** 2
**Contribution:** 2
**Rating:** 3
**Confidence:** 3

**Summary:**

This paper addresses the evolving threat of backdoor attacks in deep learning models, where hidden triggers can be covertly embedded to control model behavior at inference. Traditional attacks use external patches or perturbations as triggers, but they often face two challenges: detection by defense mechanisms and high computational costs. To overcome these, the authors draw inspiration from cognitive neuroscience, comparing model decision-making to human cognition and proposing a new approach that manipulates internal representations directly. The proposed framework, called Representation Confusion Attack (RepConfAttack), bypasses the need for external triggers by modifying inherent concepts within the model’s representation spaces. This approach enhances stealth, making the attack less detectable by standard defenses. Experimental results show that RepConfAttack is effective, achieving high success rates even against strong defenses, suggesting a novel and advanced method for conducting undetectable backdoor attacks.

**Strengths:**

1. The paper addresses a timely and important topic, focusing the security of self-supervised learning (SSL) models against backdoor attacks. The proposed RepConfAttack introduces a novel approach that leverages cognitive science to manipulate internal representations, enhancing stealth and evading detection.

2. The findings are inspiring, and the perspective using cognitive science adds an interesting dimension.

**Weaknesses:**

1. The novelty of the proposed attack lies in leveraging naturally existing concepts in the dataset, which seems similar to the existing work by Wenger et al. (NeurIPS '22). Could the authors clarify the specific technical differences from [1]?

2. While the use of cognitive science to explain the backdoor attack is intriguing, the connection feels tenuous and lacks strong motivation. First, the cognitive science aspects are described mostly in natural language without much technical depth or formalization. Second, the cognitive science section seems unnecessary, as the proposed Confusion Attack can be fully understood by reading Section 4.3 alone.

3. The practicality of the proposed attack raises some concerns. For example, if "water" is chosen as the target concept, does the attack only succeed when "water" is present in the image? How does the attack perform when this concept is absent? Traditional backdoor attacks can always succeed by injecting a trigger, ensuring consistent success.

4. Concept set size: What is the typical size of the concept set in the experiments? Additionally, how does the size of the concept set affect the performance of the proposed method?

5. Determination of threshold $\sigma$: How is $\sigma$ determined in the experiments, and how sensitive is the performance of the proposed method to variations in $\sigma$

6. The paper misses some backdoor defense strategies specifically designed for SSL, such as SSL-Cleanse[2] and DECREE[3].
Discussing how RepConfAttack could potentially evade these defenses would strengthen the paper's security analysis.


[1] Wenger et al., Finding Naturally-Occurring Physical Backdoors in Image Datasets, NeurIPS '22.

[2] Zheng et al., SSL-Cleanse: Trojan Detection and Mitigation in Self-Supervised Learning, ECCV '24.

[3] Feng et al., Detecting Backdoors in Pre-trained Encoders, CVPR '23.

**Questions:**

Please respond to each weakness mentioned above.

---

> ### Author Response · Authors · 2024-11-24
> **Reply to Reviewer 9Sod**
>
> >**W1**: The specific technical differences from [1]
>
> **A1**: The difference between our work and physical backdoor attacks lies in the use and nature of triggers. Physical backdoor attacks typically use physical triggers that exist in the real world. These triggers are designed to work across varying conditions like angles, lighting, and distances. While the trigger itself may vary in appearance due to these conditions, its essence remains consistent with traditional backdoor attacks: the model is conditioned to misbehave when this specific trigger is present, whether digital or physical. Our method manipulates the model’s behavior in a more subtle and trigger-free way. This means that there is no external cue in the input data that the model depends on to exhibit the backdoor behavior. This represents a significant conceptual difference, as we bypass the need for any inserted trigger altogether. Our approach addresses limitations inherent to traditional and physical backdoor attacks, particularly in scenarios where the presence of a trigger can be detected or mitigated.
>
> >**W2**: The cognitive science section seems unnecessary.
>
> **A2**: To enhance clarity, we revised our discussion of cognitive science, removing Section 4.1 and streamlining the introduction to focus on essential connections between the Hopfieldian view and modern explainable AI. The Hopfieldian perspective has inspired methods for understanding neural networks' learned representations—specifically, how visual features and human-interpretable concepts drive model decisions. This understanding of concept-driven neural activations provides the key motivation for our proposed attack method.
>
> >**W3**: The practicality of the proposed attack.
>
> **A3**:We have set a threshold of the target concept in the training process, and in practice, only when the concept score of the target concept is higher than the threshold, then the attack will success. We conducted additional experiments using multiple combined concepts as triggers. After the backdoor attack, if any one of these concepts in the output image exceeds its threshold, the model predicts the target label. Using mixed concepts as triggers significantly enhances the practicality of this backdoor attack. The results of the mixed concept as a trigger is shown below.
>
> **Table: Combined concepts as backdoor triggers: performance on CIFAR-10 using TCAV as concept score calculator with high ASR(%) and CACC(%)**
>
> | Concepts | CACC | ASR |
> |----------|------|-----|
> | Airplane+Oven | 94.2 | 96.7 |
> | Engine+Headlight | 95.4 | 95.5 |
> | Head+Clock | 95.6 | 93.8 |
> | Mirror+Air-conditioner | 93.4 | 95.1 |
> | Building+Cushion | 94.7 | 93.2 |
>
> >**W4**: Concept size
>
> **A4**: Our method doesn't require a fixed concept size. Our methodology is invariant to the input dimensions or concept sizes, as it operates solely on the extracted concept scores rather than the intermediate visual representations.
>
> >**W5**: Threshold σ
>
> **A5**:We calculate the target concept scores for all training samples. We select the top 1% (poisoning ratio) of samples, and use the score of the last sample in this top 1% as threshold σ. In the test set, any sample with a concept score above σ will be predicted as the target label by the model.
>
> >**W6**: Defense strategies for SSL, such as SSL-Cleanse[2] and DECREE[3]
>
> **A6**:We evaluated the effectiveness of the two proposed reference defenses against our attack. These defenses fail to even detect whether our encoder is backdoored. The results are shown below.
>
> **Backdoor Detection Methods Comparison, and "true" indicates detected backdoors, "false" indicates undetected.**
> | Concept | SSL-Cleanse | DECREE |
> |---------|-------------|---------|
> | Airplane | false | false |
> | Oven | false | false |
> | Engine | false | false |
> | Headlight | false | false |
> | Head | false | false |
> | Clock | false | false |
> | Mirror | true | false |
> | Air-conditioner | false | false |
> | Building | false | false |
> | Cushion | false | false |
>
> References:
>
> [1] Wenger et al., Finding Naturally-Occurring Physical Backdoors in Image Datasets, NeurIPS '22.
>
> [2] Zheng et al., SSL-Cleanse: Trojan Detection and Mitigation in Self-Supervised Learning, ECCV '24.
>
> [3] Feng et al., Detecting Backdoors in Pre-trained Encoders, CVPR '23.
>
> [4] Kim, Been, et al. "Interpretability beyond feature attribution: Quantitative testing with concept activation vectors (tcav)." International conference on machine learning. PMLR, 2018.
>
> [5] Bhalla, Usha, et al. "Interpreting clip with sparse linear concept embeddings (splice)." NeurIPS 2024.
>
> [6] Cheng Y A, Rodriguez I F, Chen S, et al. RTify: Aligning Deep Neural Networks with Human Behavioral Decisions. NeurIPS 2024.

---

### Official Review · Reviewer_jJUJ · 2024-11-04

**Soundness:** 2
**Presentation:** 3
**Contribution:** 2
**Rating:** 5
**Confidence:** 5

**Summary:**

This paper presents RepConfAttack, a backdoor attack framework that exploits internal concept representations within CLIP models instead of traditional external triggers, drawing inspiration from cognitive neuroscience. The method achieves high attack success rates while maintaining clean performance across multiple datasets.

**Strengths:**

- First to apply the Hopfieldian view from cognitive neuroscience to explain backdoor attack mechanisms.
- Proposes a novel concept representation-based backdoor attack method (RepConfAttack) without requiring external triggers

**Weaknesses:**

- Only validated on image classification tasks downstream of CLIP, leaving other potential tasks unexplored
- Claims to target CLIP but lacks comparison with representative CLIP-specific attacks like BadCLIP, which weakens the comparative analysis
- Insufficient justification for concept selection threshold (σ) choice
- Limited sensitivity analysis on poisoning rate
- No validation on larger-scale datasets, such as ImageNet

**Questions:**

- How to select optimal trigger concepts? Are certain concepts inherently more suitable as triggers?
- Have you considered using combinations of multiple concepts as triggers?
- Can this method be extended to other types of multimodal models?
- How does the method perform on larger-scale datasets?

---

> ### Author Response · Authors · 2024-11-24
> **Reply to Reviewer jJUJ**
>
> >**W1**：Leaving other potential tasks unexplored
>
> **A1**: This paper focuses on backdoor attacks in CLIP's classification tasks, which is consistent with most existing backdoor attacks on CLIP[1]. Thus we think focusing on classification already has significant contributions compared to those related work.
>  In future work, we plan to extend our approach to prompt generation tasks, where concepts will serve as triggers to make multimodal large models generate specific biased responses, enabling more sophisticated backdoor attacks.
>
> >**W2**：Claims to target CLIP but lacks comparison with representative CLIP-specific attacks like BadCLIP.
>
> **A2**:  In the revised version we have included BadCLIP as a baseline for comparison. Our results show that its attack performance remains inferior to our proposed attack method. Moreover, BadCLIP can be detected and defended against by various defense methods while our attack remains effective. The results of BadCLIP as a attack baseline is shown below.
> **Clean Accuracy (CACC) (%) and Attack Success Rate (ASR) (%) of BadCLIP against different kinds of defense baselines**
> | Dataset | Defense Method | w/o | ShrinkPad | Auto-encoder | SCALE_UP | FineTune | ABL |
> |---------|---------------|-----|------------|--------------|----------|----------|-----|
> | CIFAR10 | CACC | 96.2 | 93.5 | 94.2 | 95.9 | 93.7 | 89.4 |
> |  | ASR | 99.6 | 88.8 | 0.4 | 0 | 0.2 | 0 |
> | CIFAR100 | CACC | 83.3 | 80.1 | 78.6 | 78.2 | 82.2 | 77.1 |
> |  | ASR | 96.2 | 76.3 | 0.4 | 0.5 | 0 | 0.1 |
> | Tiny-Imagenet | CACC | 70.5 | 67.3 | 68.1 | 63.2 | 68.7 | 67.5 |
> |  | ASR | 87.8 | 79.2 | 1.7 | 0.9 | 0.3 | 0.6 |
>
> >**W3**：Insufficient justification for concept selection threshold (σ) choice
>
> **A3**:  If we use "water" as a trigger concept, we calculate concept scores for all samples in the training set. We select the top 1% (the poisoning ratio) of samples with the highest scores. The concept score of the last sample in this top 1% becomes our threshold σ (different concepts have different thresholds). For the test set, any sample with a concept score higher than σ will be predicted as the target label by the model.
>
> >**W4**：Limited sensitivity analysis on poisoning rate
>
> **A4**: We do not agree. We have presented the ablation experiments on different poisoning rates in Table 7 of the paper. Table 7 shows that even with reduced poisoning ratios, our attack maintains nearly 100% ASR while keeping CACC above 97%, demonstrating it effectiveness even with minimal poisoned data
>
> >**Q1**: How to select optimal trigger concepts?
>
> **A5**: Indeed, some concepts work better as triggers. For example, using "engine" as a trigger with "airplane" as the target label leads to more successful attacks. While using triggers that are very different from the original label may slightly reduce effectiveness, the attack success rate generally remains above 90%.
>
> >**Q2**: Using combinations of multiple concepts as triggers
>
> **A6**: In the revised version, we added experiments using mixed concepts as triggers. See page 8 for details. When using two concepts as triggers, the model predicts the target label if either concept's score exceeds its threshold in test data. This mixed-concept approach makes backdoor attacks more stealthy by diversifying trigger conditions, making detection and defense more challenging. This attack method could be extended to use more combined concepts for even more covert backdoor attacks. The results is shown in the Table below.
>
> **Table: Combined concepts as backdoor triggers: performance on CIFAR-10 using TCAV as concept score calculator with high ASR(%) and CACC(%)**
>
> | Concepts | CACC | ASR |
> |----------|------|-----|
> | Airplane+Oven | 94.2 | 96.7 |
> | Engine+Headlight | 95.4 | 95.5 |
> | Head+Clock | 95.6 | 93.8 |
> | Mirror+Air-conditioner | 93.4 | 95.1 |
> | Building+Cushion | 94.7 | 93.2 |
>
> >**Q3**: Can this method be extended to other types of multimodal models?
>
> **A7**: This work can definitely be extended to multimodal text generation. In MLLMs (Multimodal Large Language Models), if an input image contains certain concepts, it could trigger a backdoor that causes the model to generate text with specific biases, violent content, or jailbreak responses. This creates a subtle form of backdoor attack in the multimodal domain, where visual triggers influence text generation behavior. However, as in this paper we mainly focus on CLIP. Extending to other MLLMs will be future work.
>
> >**Q4** :How does the method perform on larger-scale datasets?
>
> **A8**: We are currently validating our attack on the Caltech-101 dataset. While ImageNet would be ideal for evaluation, its scale exceeds our GPU capacity. Results will be updated upon completion of these experiments.
>
> *References:*
>
> [1] Liang S, Zhu M, Liu A, et al. Badclip: Dual-embedding guided backdoor attack on multimodal contrastive learning[C]//Proceedings of the IEEE/CVF Conference on Computer Vision and Pattern Recognition. 2024: 24645-24654.

---

> > ### Comment · Reviewer_jJUJ · 2024-11-26
> >
> > Thank you for your detailed response. After reviewing the comments from other reviewers, I agree with their perspective that the proposed method has limited relevance to cognitive science. I also noticed that you have updated the manuscript with a new version; however, I still find some aspects of the revised explanations unclear. For instance, I could not find any reference to Figure 1 in the Introduction. Additionally, I look forward to seeing the results on the Caltech-101 dataset.

---

### Official Review · Reviewer_E6Up · 2024-11-04

**Soundness:** 2
**Presentation:** 3
**Contribution:** 3
**Rating:** 5
**Confidence:** 3

**Summary:**

This work proposes an interesting "concept-based" backdoor attack where backdoor poisoning images are labeled following the concept score from explainable AI tools. In particular, if the concept vector or a given image is larger than a pre-defined threshold, the label of the given image will be changed to the target label. After training on the poisoned data set, images with an "internal trigger" (i.e., concept vector larger than the threshold) will be wrongly classified as the target class. The authors also motivate the design choice with the cognitive neuroscience theory. The proposed attack is claimed to be stealthy against existing baseline defenses.

**Strengths:**

This paper is well-structured, with detailed explanations and a background provided to support each step of the attack design. The authors share an interesting observation that the "concept activation" is the driving force behind the backdoor attacks. Extensive experiments are also provided to validate the proposed attack's effectiveness, where the attack is especially effective against backdoor defenses.

**Weaknesses:**

- This attack jumps out of the regular threat model where an adversary puts the trigger on test samples during inference to manipulate the predictions. This change makes the backdoor attack more stealthy since there is no need to add external triggers during the test. However, it also brings a question about how to implement it in practice when the adversary needs to change the prediction on a certain target images. For example, the concept vector threshold of the target image is indeed lower than the pre-defined value. Is there a feasible way to increase the value to trigger the backdoored model? One possible solution is to deliberately change the concept vector score of the target image. It would be great if the authors could clarify this point.

- Related to the question above, if the adversary manages to change the concept vector score, the proposed method is similar to a regular backdoor-triggered sample. In this case, how reliable is the concept score? Taking the results of table 2 also into consideration, when does the proposed attack fail, and is it caused by the concept score calculation? In addition, can the same strategy be used in feature space for random statistics instead of calculating the concept vector score?

- Adaptive defenses. One major advantage of the proposed attack is that it can resist backdoor defense. Regarding the working mechanism of the previous defense, it is non-surprise that previous defenses do not work for the proposed attack. For example, ShrinkPad must inspect the difference between clean and triggered images. The proposed attack is somehow unrelated to ShrinkPad since the triggered inputs do not include substantial patterns. Given the existing defenses, would an adaptive defense that considers the proposed attack mechanism easily filter the proposed backdoor attack? It would be great if the authors could clarify this point.

Minor:
Line 219, bold -> green.
The font size in Figure 4 is too small.

**Questions:**

Please clarify about triggering on a randomly selected test image, the reliability of the concept score, and adaptive defenses.

---

> ### Author Response · Authors · 2024-11-24
> **Reply to Reviewer E6Up**
>
> >**W1**: How to implement it in practice
>
> **A1**: Since the target image has many concepts, our method supports composite concepts with conjunctions, i.e., the attack will succeed if any one concept is higher than the threshold. Thus, if the concept vector threshold of the target image is indeed lower than the pre-defined value, we can use composite concepts, i.e., find the target image with one concept higher threshold, to make the attack effective. As we mention in the introduction, the internal states of image classifiers can be represented as human-understandable concepts. Thus, it is impossible for all concepts to be lower than the threshold, as if all concepts/representations are irrelevant to the class, then this image should not belong to the specific class.
>
> We appreciate your suggestion to deliberately alter the concept vector score of the target image, but we want to clarify that the concept score is an inherent property of the images and the classifier and cannot be changed directly. This aligns with our notion that concepts are intrinsic to the neural network’s understanding and must be activated rather than modified.
>
> Additionally, we emphasize that our scenario is more practical, as the targeted label has a strong relationship with the “explicit concept,” making it more intuitive and human-understandable. For example, both ducks and swans are typically associated with water, so misclassifying a duck as a swan appears more natural. In contrast, the traditional approach of misclassifying a duck as a tiger, while stealthy, is less intuitive and more confusing.
>
> >**W2.1**: How reliable is the concept score?
>
> **A2.1**: Firstly, we respectfully argue that the adversary CANNOT change the concept vector scores of any images since these concept scores are inherent properties of the corresponding images and the networks. Specifically, for a given image and its concept vector, a high score of a specific concept means that this image contains such a concept and vice versa. Since our backdoor attack would not modify any training images, the adversary thus could not change any concept vector scores.
>
> Secondly, we note that the concept score is reliable across different concept extraction methods through extensive experiments in Table 8 in the paper.
>
> >**W2.2**: When does the proposed attack fail, and is it caused by the concept score calculation?
>
> **A2.2**: We clarify our proposed attack will fail ONLY in the case where all training and targeted images do not contain any concept from our pre-selected concept set $C$. However, such a case is almost impossible to happen in real-world threat scenarios since to make the attack effective, the adversary will select a concept set to cover concepts in real-world images as much as possible.
>
> >**W2.3**:Can the same strategy be used in feature space for random statistics instead of calculating the concept vector score?
>
> **A2.3**: Our strategy cannot adopt concept vectors calculated from random feature spaces. Specifically, concepts are natural features that align well with human perception. The reason that our proposed backdoor attack can work is that it leverages these natural features (i.e., concepts) that widely exist in real-world images. Unfortunately, these natural features apparently do not exist in random feature spaces, thus concept vectors calculated from such spaces would not have meaningful information, and our backdoor attack could not leverage such meaningless concept vectors.
>
> >**W3**: Adaptive defenses.
>
> **A3**: We further evaluated our backdoor attack against two recently proposed defense methods: SSL-Cleanse[1] and DECREE[2]. Our results show that both defense methods fail to detect whether the encoder has been backdoored with the target concept, let alone remove the backdoor effectively. The result is shown in the Table below.
>
> **Backdoor Detection Methods Comparison, and "true" indicates detected backdoors, "false" indicates undetected.**
> | Concept | SSL-Cleanse | DECREE |
> |---------|-------------|---------|
> | Airplane | false | false |
> | Oven | false | false |
> | Engine | false | false |
> | Headlight | false | false |
> | Head | false | false |
> | Clock | false | false |
> | Mirror | true | false |
> | Air-conditioner | false | false |
> | Building | false | false |
> | Cushion | false | false |
>
> For the adaptive defenses, note that this is the paper introducing confusing the concept vectors as attacks, we think it is a good question but it is beyond the scope of the paper.
>
> >**Minor**: Minor: Line 219, bold -> green. The font size in Figure 4 is too small.
>
> **A4**: Thanks & addressed.
>
> >**Questions**: Please clarify about triggering on a randomly selected test image, the reliability of the concept score, and adaptive defenses.
>
> **A5**: Please refer to the replies above
>
> References:
>
> [1] Zheng et al., SSL-Cleanse: Trojan Detection and Mitigation in Self-Supervised Learning, ECCV '24.
>
> [2] Feng et al., Detecting Backdoors in Pre-trained Encoders, CVPR '23.

---

> > ### Comment · Reviewer_E6Up · 2024-12-03
> >
> > Dear authors, thank you for your detailed analysis and clarification. This paper discussed indeed an interesting threat, but it’s hard to attribute the new method to backdoor  attacks from my understanding. I would suggest the authors to position the proposed threat in a more accurate way and provide more discussions on the concept score especially under more challenging scenarios, e.g., adversarial examples.  I would maintain my original recommendation.

---

### Author Response · Authors · 2024-11-24
**General Response**

**For writing**

We have revised the introduction and Section 4 regarding cognitive neuroscience and the Hopfieldian View. We deleted Section 4.1 (preliminary of cognitive science) as it is less related to our method. We also shorten and weaken the cognitive science part in the introduction, and we add more intuitions on the connection between the Hopfieldian view and current concept extraction methods in explainable AI. We also declared the differences between the physical backdoor[2] and our attack method.

**For experiments**

We added BadCLIP[1] as an attack baseline, demonstrating that its attack performance is inferior to our method. We evaluated two backdoor detection methods, SSL-Cleanse[3] and DECREE[4], against our attack and found that neither could detect our backdoored encoder. Additionally, we explored using pairs of combined concepts as backdoor triggers, which maintained strong attack performance.

References：

[1] Bai, Jiawang, et al. "BadCLIP: Trigger-Aware Prompt Learning for Backdoor Attacks on CLIP." Proceedings of the IEEE/CVF Conference on Computer Vision and Pattern Recognition. 2024.

[2] Backdoor Attacks Against Deep Learning Systems in the Physical World.

[3] Zheng et al., SSL-Cleanse: Trojan Detection and Mitigation in Self-Supervised Learning, ECCV '24.

---

### Meta-Review · Area_Chair_e9kZ · 2024-12-19

**Metareview:**

This work studied the backdoor attack against CLIP model, and proposed a Representation Confusion attack method which picks the image containing particular concepts as the poisoned image and changes its label to the target label. Compared with most existing backdoor attacks, this method doesn't need to design and insert a trigger into the poisoned image.

It received 4 detailed and professional reviews. Most reviewers recognized that the cognitive science inspired backdoor attack provides a new perspective, and the observations on backdoor attack presented in Sec. 4 are also interesting.

Meanwhile, there are also several important concerns, mainly including:
1. The novelty is limited, since it is similar with some existing works, such as natural backdoor, physical backdoor.
2. The methodology design is weakly relevant to cognitive science.
3. Experiments: some existing backdoor attacks against CLIP are not compared, such as BadCLIP; the defense methods are outdates, some latest defenses like VDC are not adopted, and some adaptive defenses are not evaluated; the evaluated tasks and datasets are narrow.
4. The studied threat model may not belong to backdoor attack (mentioned by one reviewer).

There are sufficient discussions between authors and reviewers. My judgements about above points are as follows:
1. The proposed concept based trigger is indeed not new, as the exactly same idea has been studied in very old backdoor attacks. For example, in [1], the image containing some objects with particular attributes (e.g., green car) is chosen as the poisoned image. It is also similar in [2]. Meanwhile, since the semantic object serves as the trigger, it is very suitable for physical attack, and I think that is why two reviewers mentioned the similarity to physical attack.
2. My feeling is that the introduced cognitive science and the presented method are indeed somewhat weakly relevant. But I don't consider it is a significant drawback. Exploring new perspectives and introducing new inspirations are encouraged.
3. The authors added some results, such as BadCLIP and two adaptive defenses. However, they refused to try the VDC defense, and the reason is unconvincing to me, and I believe VDC could successfully defend the proposed attack, as there are semantic inconsistency between the image and its label. More downstream tasks and datasets are not added.
4. I think this work still belongs to backdoor attack, though it didn't insert one trigger into the image, but there is still trigger.

In summary, point 1 (novelty) and point 4 (insufficient experiments) are the main reasons for the rejection recommendation.

[1] How To Backdoor Federated Learning. AISTATS 2020.
[2] Finding Naturally Occurring Physical Backdoors in Image Datasets, NeurIPS 2022.

**Additional Comments On Reviewer Discussion:**

The rebuttal and discussions, as well as their influences in the decision, have been summarized in the above metareview.

---

### Decision · Program_Chairs · 2025-01-22

Reject